# Analysis of the NK2 homeobox gene *ceh-24* reveals sublateral motor neuron control of left-right turning during sleep

**Juliane Schwarz, Henrik Bringmann\***

Max Planck Institute for Biophysical Chemistry, Göttingen, Germany

**Abstract** Sleep is a behavior that is found in all animals that have a nervous system and that have been studied carefully. In *Caenorhabditis elegans* larvae, sleep is associated with a turning behavior, called flipping, in which animals rotate 180° about their longitudinal axis. However, the molecular and neural substrates of this enigmatic behavior are not known. Here, we identified the conserved NK-2 homeobox gene *ceh-24* to be crucially required for flipping. *ceh-24* is required for the formation of processes and for cholinergic function of sublateral motor neurons, which separately innervate the four body muscle quadrants. Knockdown of cholinergic function in a subset of these sublateral neurons, the SIAs, abolishes flipping. The SIAs depolarize during flipping and their optogenetic activation induces flipping in a fraction of events. Thus, we identified the sublateral SIA neurons to control the three-dimensional movements of flipping. These neurons may also control other types of motion.

## Introduction

Sleep is a widespread phenomenon found in all animals that have a nervous system and that have been studied carefully. It is defined by behavioral criteria and can be identified in animals by the lack of voluntary movements, assumption of a specific posture, an increased arousal threshold, and homeostatic regulation (*Campbell and Tobler, 1984*; *Cirelli and Tononi, 2008*; *Allada and Siegel, 2008*). While sleep appears to be a relatively inactive state, at least if seen from a behavioral point of view, it is essential and likely serves several functions ranging from optimizing behavior, basic control of energy metabolism, macromolecule synthesis and clearance, to memory consolidation and gaining of deeper insight into logical problems (*Cirelli and Tononi, 2008*; *Diekelmann and Born, 2010*; *Stickgold and Walker, 2004*; *Stickgold, 2012*; *Mackiewicz et al., 2007*; *Siegel, 2009*; *Xie et al., 2013*). Because of its importance, sleep is under homeostatic control that ensures that enough sleep takes place (*Borbély, 1982*; *Porkka-Heiskanen, 2013*). The profound physiological and behavioral changes during sleep are controlled by the nervous system. Central to the control of sleep are sleep-active sleep-promoting neurons that release inhibitory neurotransmitters, such as GABA and neuropeptides, at sleep onset (*Saper et al., 2005*, *2010*). Genetic analysis of sleep in different model organisms has uncovered sleep regulatory mechanisms and has shown that many pathways have conserved functions across species, supporting the view that sleep is evolutionarily ancient and conserved (*Sehgal and Mignot, 2011*; *Singh et al., 2014*; *Zimmerman et al., 2008*).

*C. elegans* is an established model system to study the molecular basis of behavior. The hermaphrodite has a small nervous system containing 302 neurons with a known and invariant connectivity. The short generation time and molecular accessibility make it an attractive model to study (*Brenner, 1974*; *White et al., 1986*). At the end of each larval cycle, *C. elegans* molt. Before shedding the old cuticle, larvae go through a developmental stage and behavioral state called lethargus during which they don't feed. Locomotion behavior during lethargus can be described as quiescence

**\*For correspondence:** henrik.
bringmann@mpibpc.mpg.de

**Competing interests:** The authors declare that no competing interests exist.

**eLife digest** Although sleeping individuals do not move voluntarily, they are not completely immobile. Both people and animals regularly change position in their sleep, but it is not known why these movements occur or what regulates them.

One of the simplest animals known to require sleep is the nematode worm *Caenorhabditis elegans*, which is often used by researchers to study the molecular basis of behavior. In common with more complex animals, worms go to sleep lying on either their left or right side and then switch periodically between the two. This "flipping" behavior is typically not seen outside of sleep.

By screening worms with mutations in different genes, Schwarz and Bringmann identified one mutant that does not flip during sleep. The mutant lacked a gene called *ceh-24*, which is normally active in a set of four neurons known as SIAs. These are a type of motor neuron; that is, neurons that control the contraction of muscles.

The body wall muscles of *C. elegans* run along the length of its body and are organized into "quadrants" that each cover a quarter of the worm. Schwarz and Bringmann show that unlike other *C. elegans* motor neurons, SIA neurons control each quadrant separately. By activating specific SIA neurons the worms can contract the muscles on each side of the body independently, and thereby flip from one side to the other.

Further investigation revealed that the SIA motor neurons can also control other types of complex movement. Additional experiments are now needed to determine how the neurons support these behaviors. Another challenge will be to work out the purpose of posture changes during sleep for *C. elegans* and other animals.

bouts that are intermitted by motion bouts (*Iwanir et al., 2013*; *Nagy et al., 2014a*, *2014b*). Developmentally controlled quiescence behavior fulfills the behavioral criteria that define sleep in other organisms, such as decreased voluntary movement, an increased arousal threshold, reversibility, and homeostatic regulation, and is thus called sleep (*Iwanir et al., 2013*; *Nagy et al., 2014a*; *Raizen et al., 2008*; *Trojanowski et al., 2015*; *Trojanowski and Raizen, 2016*; *Driver et al., 2013*; *Schwarz et al., 2011*, *2012*; *Turek et al., 2016*; *Cassada and Russell, 1975*). *C. elegans* larvae display behavioral quiescence also during dauer diapause (*Cassada and Russell, 1975*; *Gaglia and Kenyon, 2009*). In the adult, stress, satiety, and reduced insulin signaling can induce behavioral quiescence (*Gaglia and Kenyon, 2009*; *Van Buskirk and Sternberg, 2007*; *Hill et al., 2014*; *You et al., 2008*). How these different types of quiescence are related and whether all of them fulfill the definitions for sleep currently is under investigation. It appears that different types of quiescence are controlled by both distinct and overlapping mechanisms (*Trojanowski et al., 2015*; *Trojanowski and Raizen, 2016*; *Kayser and Biron, 2016*). Here we study developmentally controlled sleep behavior in the larva.

The timing of lethargus is controlled by LIN-42, a homolog of the circadian regulator PER, that oscillates with the sleep-wake and developmental rhythm (*Jeon et al., 1999*; *Monsalve et al., 2011*). Crucial to the induction of larval sleep behavior during lethargus is a single neuron called RIS, a GABAergic and peptidergic neuron, which depolarizes at the onset of sleep and actively induces sleep by releasing inhibitory neurotransmitters including FLP-11 peptides (*Turek et al., 2016*, *2013*). Peptide release to control behavioral states appears to be a common theme in sleep regulation in various species (*Turek et al., 2016*; *Richter et al., 2014*; *Nelson et al., 2013*, *2014*). Thus, RIS and mammalian sleep-active neurons are functionally similar, because both are active at sleep onset and actively induce sleep through inhibitory transmitters (*Saper et al., 2005*).

*C. elegans* is typically cultured on the flat surface of agarose plates where it lies on its side and propels itself by 'crawling', a locomotion behavior consisting of undulating body movements created by sinusoidal dorsal and ventral muscle contraction. Most motor neurons that control undulating body movement are found in the ventral cord. These neurons innervate both dorsal and ventral muscle. The body wall muscles run along the long axis of the body and are organized into four quadrants, two dorsal and two ventral quadrants, which run along the side of the dorsal cord or the ventral cord, respectively. Separate groups of cholinergic motor neurons exist that drive forward or

backward movement through excitatory neuromuscular junctions onto dorsal or ventral muscle. In addition, GABAergic motor neurons play a modulatory, facilitating role in movement through contra-lateral inhibition. Thus, the motor circuits for crawling on plane surfaces are relatively well defined (*Zhen and Samuel, 2015*). In contrast, natural environments are rugged three-dimensional land-scapes through which animals have to navigate. Simple dorsal and ventral contractions can hardly explain three-dimensional movements, which would require additional left-right body movements and thus a more complex regulation of musculature though a separate innervation of each muscle quadrant. *C. elegans* shows a number of additional types of movements that appear to be more complex than locomotion on a plane surface such as burrowing into substrates, nictation, which is a dispersal behavior during which worms assume an upright posture, and turning from their left to their right side (*Lee et al., 2012*; *Beron et al., 2015*; *Vidal-Gadea et al., 2015*). Left-right turning, or flipping, occurs mostly during lethargus. It is a movement in which the worm turns 180° around its longitudinal axis. It was suggested that this behavior might help loosen the old cuticle, but causality was never tested. Flipping does not normally occur outside of lethargus except after stressful han-dling of the worms (*Singh and Sulston, 1978*). Most flips were shown to occur during quiescence bouts but they can also be found during motion bouts. They are usually initiated in worms that con-tain a relaxed body posture that contains only one body bend, which is the typical posture during lethargus. By contrast, worms outside of lethargus typically contain more than one body bend. The emergence of flips from a single-bend posture suggests facilitation by this posture, consistent with the idea that biomechanical constraints exist for this behavior. Thus, hypothetically, flipping occurs mostly during lethargus because this stage is characterized by reduced body bends. Flipping is asso-ciated with sleep behavior in the sense that most flips emerge from quiescence bouts during which there is a sleep-specific, relaxed, posture (*Tramm et al., 2014*). Even though flipping behavior is long known, the molecular and neural substrates and molecular mechanisms through which worms flip are unknown.

Here, we study left-right turning during larval sleep in *C. elegans* as an example for a sleep-asso-ciated behavior. We identified mutants in the NK2 homeobox gene *ceh-24* that do not flip during sleep. Analysis of *ceh-24* combined with calcium imaging and optogenetics suggests that the sublat-eral SIA neurons are required for flipping but also for other types of movements. The SIAs appear to act as motor neurons that innervate all of the four muscle quadrants separately, and they are thus ideally suited to control complex movements such as flipping.

## Results

### Flipping is associated with sleep behavior

Flips have been reported to occur mostly during lethargus quiescence bouts (*Tramm et al., 2014*). To corroborate these findings, we looked at wild type worms and analyzed their locomotion speed before flipping. We cultured L1 larvae in microfluidic chambers that were filled with bacterial food and filmed their development and behavior from the mid L1 stage until the early L2 larva using time-lapse video microscopy (*Bringmann, 2011*; *Turek et al., 2015*). We identified lethargus in these movies by the absence of feeding as seen by the cessation of pharyngeal pumping and manually scored the videos for flipping by monitoring the side on which the worms were lying by identifying the position of the developing gonad, which is located ventrally (*Figure 1A*). We measured the movement speed in the five seconds before each flip and plotted a frequency distribution of speeds versus the frequency distribution of flipping. The speed distribution showed that most flips occurred during low movement speeds, consistent with the idea that flips emerge from quiescence bouts (*Figure 1B*) (*Tramm et al., 2014*). To test whether sleep behavior only coincides with or actually is required for flipping we analyzed a mutant that has greatly impaired sleep. We used a deletion in *aptf-1*, a conserved regulator of sleep, which completely eliminates locomotion quiescence during lethargus, and counted flips in this mutant (*Turek et al., 2013*; *Kucherenko et al., 2016*). Mutant worms went through the molting cycle, but flipping during lethargus was strongly reduced in *aptf-1 (-)*, consistent with the idea that sleep facilitates flipping (*Figure 1C*). Thus, our results support the view that most flips occur during periods of low mobility that are characteristic of sleep. Also, sleep-ing behavior appears to facilitate flipping.

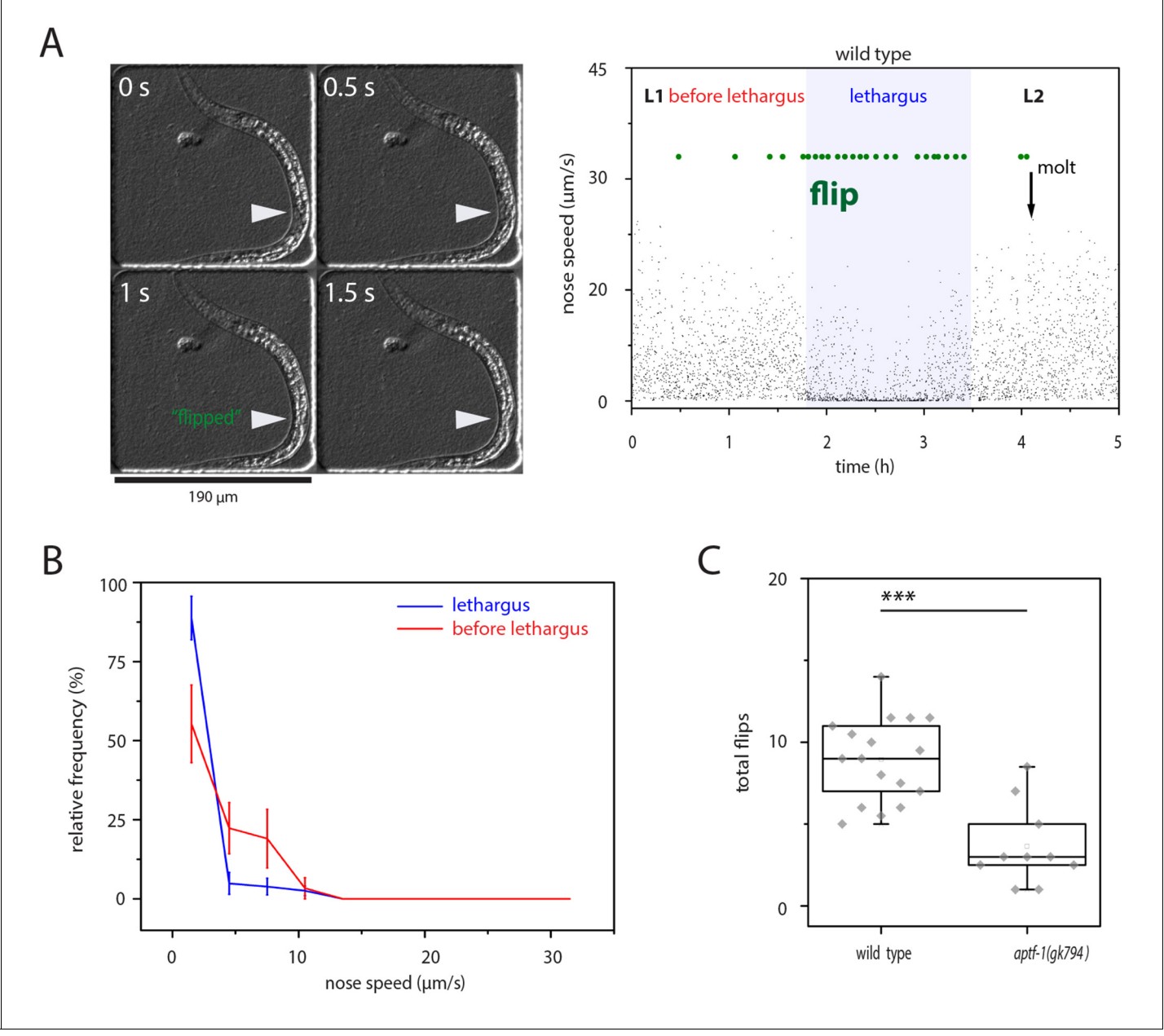

**Figure 1.** left-right turning is associated with sleep and appears to be facilitated by this behavior. (A) Flipping during sleep: Images showing flipping in an example time lapse movie of an L1 larva cultured in a hydrogel microcompartment. Shown are DIC images. At time point 0s the developing gonad is located on the left side (arrowhead), at time point 0.5 s the larva turns and at time points 1s and 1.5s the developing gonad can be seen on the right side. The right plot shows the behavior over time during the L1 to L2 stage for one animal. Lethargus is defined here as a lack of feeding as seen as a lack of pharyngeal pumping and it is the phase in which sleep occurs, i.e. movement is strongly reduced and quiescence bouts occur. Nose speed measurements show the reduction of mobility during sleep. Flips are displayed in green. (B) Most flips occur during phases of low mobility. Shown is a frequency distribution of flips as a function of movement five seconds before the flip. Compare also images in A). (C) Flipping is strongly reduced in *aptf-1(-)*, which is lacking sleep behavior, consistent with the view that sleep behavior facilitates flipping.

## *ceh-24* is required for flipping

No mutants have been reported that have normal locomotion quiescence but lack flipping behavior. In order to dissect sleep-specific flipping behavior, we decided to identify such mutants. We looked through existing movies that we had taken previously in a screen to identify sleep mutants. For the

initial screen, mutant strains were cultured inside agarose hydrogel microchambers and were filmed and scored for flipping as described above. We found one strain that appeared not to flip (methods). This strain carried a deletion in *ceh-24.* To characterize the *ceh-24* mutant phenotype, we cultured worms again inside microfluidic chambers made from agarose hydrogel and quantified both locomotion quiescence and flipping behavior in two *ceh-24* mutants. The *ceh-24(cc539)* allele identified in the screen deletes almost the entire gene and likely is a molecular null. *ceh-24(tm1103)* is an independently isolated allele that also deletes large parts of the gene. Lethargus was clearly identifiable in the wild type and in the mutants by a phase of feeding cessation as seen by the absence of pharyngeal pumping, which was followed by cuticle shedding. Both mutants showed strongly reduced flipping behavior. Whereas wild type worms flipped 0.4/h during wake and 6.4/h during sleep, both *ceh-24* mutants had strongly reduced flipping, 0/h during wake and 0.1/h during sleep, with most individuals showing no flipping at all (*Figure 2A*). The presence of a flipping defect in two independently isolated strains carrying *ceh-24* deletions strongly suggested that *ceh-24* is required for flipping. To confirm that *ceh-24* is required for flipping we performed a rescue analysis by introducing a transgene carrying the *ceh-24* gene driven by its own promoter into a *ceh-24* deletion background. The *ceh-24* transgene rescued flipping confirming that the flipping defect seen in *ceh-24* mutants was indeed caused by deletion of this gene (*Figure 2C*). Locomotion quiescence during sleep appeared normal in *ceh-24(-)* worms as judged from an analysis of locomotion speeds and extraction of quiescence bouts. Speeds were normal and quiescence bout duration and frequency were indistinguishable in wild type and *ceh-24(-)* larvae (*Figure 2B*, *Figure 2—figure supplement 1*). Thus, we conclude that *ceh-24* is required for flipping.

Flipping has been suggested to be required for cuticle loosening and shedding. We hence measured the length of the nonpumping period and the time it took to fully shed the cuticle. Pharyngeal pumping and resumption could be scored from DIC movies of worms grown in microfluidic chambers. Typically, after pumping restarted, the worm crawled out of the old cuticle, which could also be scored manually from the same DIC movies. The time point at which the worm had crawled out of the cuticle completely, i.e. also the tail had completely left the cuticle, was scored as the completion of the molt. Despite the absence of flipping, the time the animals took from resuming pumping until complete shedding of the old cuticle was not significantly altered in *ceh-24* mutant animals and we could not detect any obvious defects in the new cuticle (*Figure 2D*). Thus, we could not find any contribution of flipping to the molting process, suggesting that flipping rather is related to other processes.

Is *ceh-24* also required for other behaviors or is it specifically required for flipping? To test for other defects in movement we looked at larvae in microchambers and at adults on a planar and on a structured surface. Compared with the wild type, *ceh-24(-)* L1 larvae grown in microchambers showed slightly slowed down nose movement (*Figure 2—figure supplement 1*). Adult mutant worms crawling on a planar NGM plate moved with normal speed and showed a normal fraction of forward and reverse movement (*Figure 2—figure supplement 2A–B*). In order to challenge the worms we assayed the ability of adults to escape from an indentation in the agar surface. We cast agar surfaces containing small box-shaped indentations by using PDMS molds and placed individual worms into these indentations. The worms then quickly crawled out of the indentation. We measured the time each individual worm needed to escape. Wild type worms pushed their heads over the rim of the indentation and crawled out without any signs of effort in about two minutes. *ceh-24* mutant worms could also eventually escape from the indentation, but needed four times longer for this (*Figure 2E*). Thus, *ceh-24(-)* had problems crawling out of an indentation. This is likely not caused by a defect in sensory systems, because *ceh-24(-)* have normal chemotaxis to both volatile and non-volatile odorants (*Harfe and Fire, 1998*). Also, *ceh-24(-)* responded normally to mechanical stimulation (*Figure 2—figure supplement 2C*). These results suggest that *ceh-24* mutant worms have defects not only in flipping but also in other types of movements.

## *ceh-24* is required for sublateral process formation and cholinergic function

Previous work showed that *ceh-24* is expressed in both muscle (m8 muscle of the pharynx and vulva) and in three types of sublateral motor neurons including the two SMDV neurons, the four SIB neurons, and the four SIA neurons (*Harfe and Fire, 1998*; *Kennerdell et al., 2009*). *ceh-24*-expressing sublateral neurons are cholinergic and they are still present in *ceh-24* mutant animals (*Harfe and*

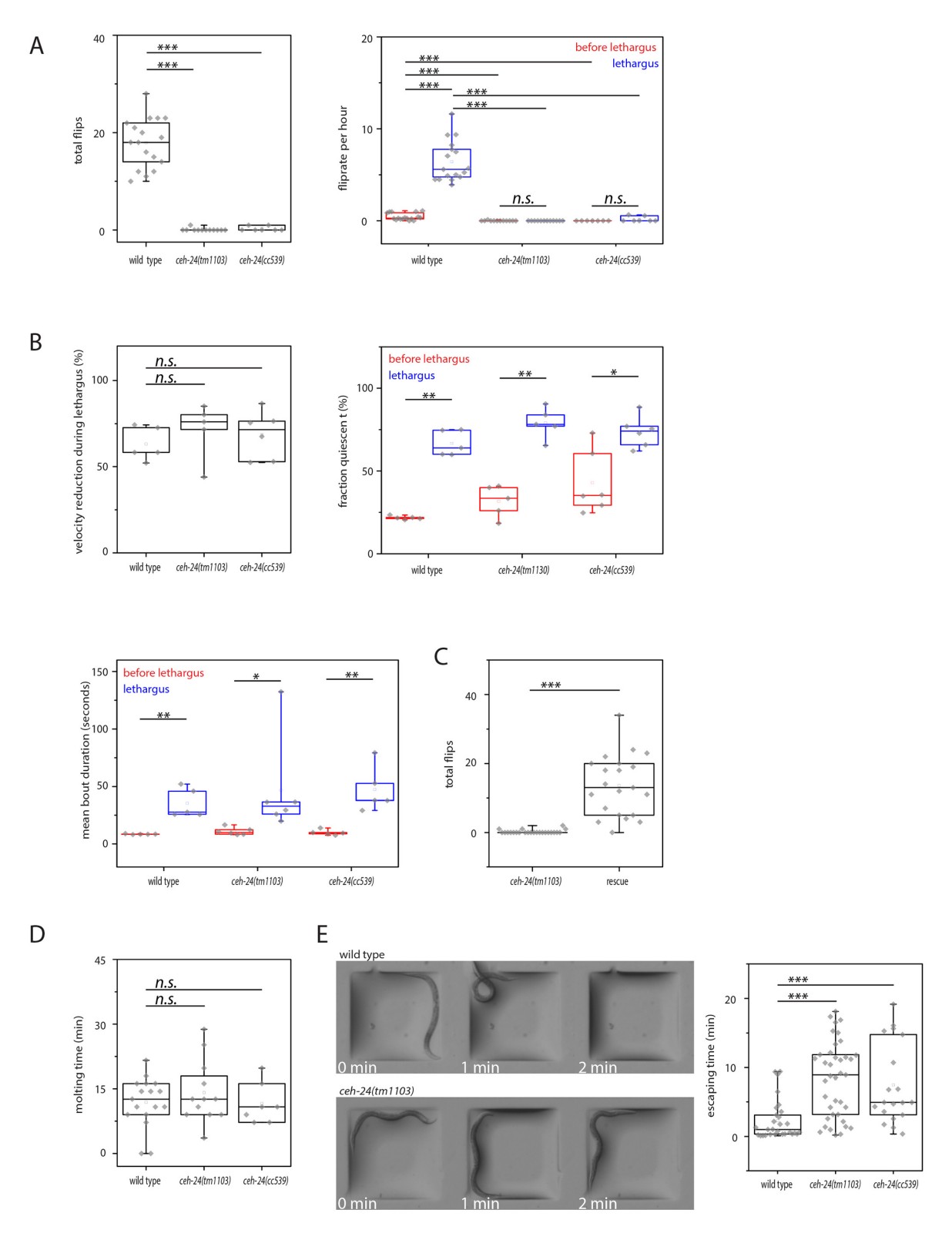

**Figure 2.** *ceh-24* is required for left-right turning and escape from an indentation. (**A**) Flipping is strongly reduced in *ceh-24(-)*: The left plot shows the reduction of total number of flips scored in a movie that lasted from two hours before sleep onset until two hours after sleep end (Kolmogorov Smirnov test, wild type [N = 17], *ceh-24(tm1103)* [N = 12], *ceh-24(cc539)* [N = 7]). The right plot shows the flip rate per hour during and outside of lethargus (paired Wilcoxon Signed Ranks test for comparison of sleep versus wake, and Kolmogorov Smirnov test for comparisons between genotypes, wild type

*Figure 2 continued on next page*

*Figure 2 continued*

[N = 17], *ceh-24(tm1103)* [N = 12], *ceh-24(cc539)* [N = 7]). (B) Sleep behavior appears normal in *ceh-24* mutants: Shown is the reduction of nose speed during lethargus compared with the speed outside of lethargus in percent (Kolmogorov Smirnov test, wild type [N = 5], *ceh-24(tm1103)* [N = 6], *ceh-24 (cc539)* [N = 6]). Also shown are quiescence bouts displayed as the fraction of worms that was quiescent and the mean quiescence bout duration. (C) A *ceh-24* transgene array rescues the flipping defect in *ceh-24(-)*: a rescue transgene containing of the *ceh-24* promoter and the *ceh-24* coding region was transformed into *ceh-24(-)* mutant worms and the total number of flips was scored (Kolmogorov Smirnov test, animals were selected that expressed the array and had normal-looking sublateral neurons with proper sublateral processes, *ceh-24(tm1103)* [N = 24], rescue [N = 21]). (D) Cuticle shedding appears normal in *ceh-24* mutant worms as judged by the time from resumption of pumping until completion of cuticle shedding (Kolmogorov Smirnov test, wild type [N = 17], *ceh-24(tm1103)* [N = 12], *ceh-24(cc539)* [N = 7]). (E) Escape from an indentation is impaired in *ceh-24(-)*: Young adult worms were placed into shallow indentations that were printed into agarose and the time the worms needed to crawl out was scored (two-sample t-test, wild type [N = 29], *ceh-24(tm1103)* [N = 39], *ceh-24(cc539)* [N = 19]). Chamber dimension was 700 μm × 700 μm, 65 μm deep. *** denotes p<0.001, n.s. denotes p>0.05.

The following figure supplements are available for figure 2:

**Figure supplement 1.** Sleep behavior appears normal in *ceh-24(-)*.

**Figure supplement 2.** Locomotion and response to mechanical stimulation of adults on an agar surface appears normal in *ceh-24(-)*.

*Fire, 1998*; *Duerr et al., 2008*; *Pereira et al., 2015*). The molecular functions of *ceh-24*, however, are unknown. We confirmed the expression pattern of *ceh-24* with a transgenic insertion that drove mKate2 expression from the *ceh-24* promoter (*Figure 3A*). To find out how *ceh-24* acts we looked at the morphology of the sublateral neurons and at their neurotransmitter function in *ceh-24* mutant worms. We crossed a *ceh-24* deletion into the fluorescence reporter driven by the *ceh-24* promoter and imaged the sublateral processes using spinning disc microscopy. In the wild type, straight sublateral processes were found to run along the muscle quadrants. In *ceh-24* mutant worms, expression from the *ceh-24* promoter was increased. Processes did not run straight along the muscle but diverted from it or branched. Processes also ended prematurely and did not extend as far posteriorly as in the wild type (*Figure 3B*). Cholinergic function is determined by the expression of two genes, *unc-17*, a synaptic vesicle acetylcholine transporter and *cha-1*, a choline acetyltransferase gene that is required for the biosynthesis of acetylcholine. *unc-17* and *cha-1* are co-transcribed in an operon. Expression of this operon can be assayed by using an *unc-17* promoter reporter, which detected cholinergic function in 159 out of 302 neurons including the SMD, SIB, and SIA neurons (*Duerr et al., 2008*; *Pereira et al., 2015*; *Rand and Russell, 1984*; *Alfonso et al., 1994*, *1993*; *Rand, 2007*). To assay for cholinergic function, we looked at the expression of a GFP reporter driven from the *unc-17* promoter. To clearly identify sublateral neurons among the many *unc-17*-expressing neurons, we also crossed a red fluorescent mKate2 marker driven by the *ceh-24* promoter into this strain. In the wild type, expression from the *cha-1* promoter was seen in SIA, SIB, and SMD neurons. However, in *ceh-24* mutant worms, expression was either undetectable or greatly reduced in all the sublateral neurons tested (*Figure 3C*). Thus, *ceh-24* is required for normal process morphology and cholinergic transmitter expression in SIA, SIB, and SMD.

Which of the cellular defects prevented flipping? Defects in sublateral process formation or lack of *unc-17* expression? Mutation of *ceh-17*, a paired-like homeobox gene, also compromises process outgrowth in SIA neurons in a way that resembles the defects we observed in *ceh-24* mutant worms (*Pujol et al., 2000*). We hence tested if *ceh-17* mutant worms were also flipping defective and whether they were also defective in *unc-17* expression of the SIA neurons. In *ceh-17* mutant worms, flipping was reduced by about half and in contrast to *ceh-24* mutant worms they had no reduction in *unc-17* expression in the SIA neurons (*Figure 3D*). This experiment suggests that the proper process morphology of the SIA neurons is at least partially required for flipping and we tested the role of cholinergic function next.

## Cholinergic function in the SIA neurons is required for flipping

The sublateral processes lie inside a thin layer of hypodermis close to and running along the body wall muscle. Each of the sublateral neurons extends one sublateral process into one muscle quadrant. The sublateral processes contain periodic swellings containing synaptic vesicles and presynaptic

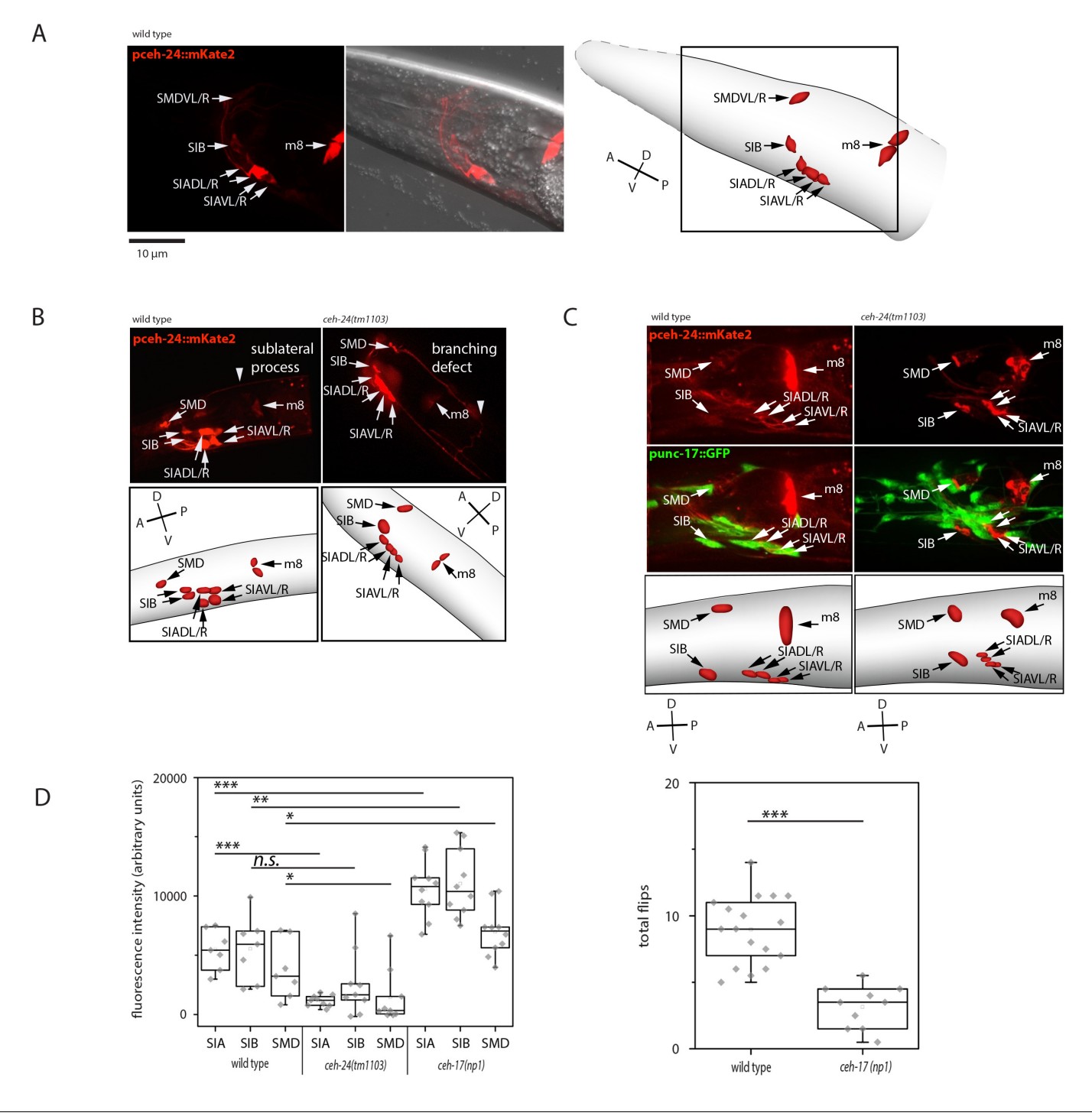

**Figure 3.** *ceh-24* is required for sublateral process formation and cholinergic function. (**A**) *ceh-24* is expressed in the sublateral SMB, SIB, and SIA neurons as well as in the m8 muscle cells: An adult animal that is expressing mKate2 from the *ceh-24* promoter is shown in DIC and in fluorescence imaging. Cartoons show worm and cell body outlines. (**B**) Sublateral processes are not formed properly in *ceh-24(-)*: processes ended prematurely and branched in *ceh-24(-)* in all animals tested (Kolmogorov Smirnov test, p<0.001) (N = 10, 10/10 wt animals had no branching and 10/10 mutant animals had branching defects). (**C**) Expression of the synaptic vesicle acetylcholine transporter gene *unc-17* is abolished or strongly reduced in the SIA neurons in *ceh-24(-)*. The plot shows the quantification of *unc-17*-expressing neurons (Kolmogorov Smirnov test, wild type [N = 7], *ceh-24(tm1103)* [N = 9]). (**D**) *ceh-17* mutants, which have a branching defect in the SIAs, have normal *unc-17* expression and have a flip defect (Kolmogorov Smirnov test, wild type [N = 7], *ceh-17(np1)* [N = 14]). *** denotes p<0.001, ** denotes p<0.01, * denotes p<0.05, n.s. denotes p>0.05.

densities suggesting that the sublateral neurons are the motor neurons that innervate adjacent muscle quadrants. The innervation pattern suggests that an individual control of each muscle quadrant by the sublateral neurons would be possible (*White et al., 1986*; *Altun and Hall, 2011*). Our results above suggest that *ceh-24* is required for flipping not only because it controls process outgrowth but also because it specifies a cholinergic transmitter expression. To test the hypothesis that cholinergic function in sublateral neurons is required for flipping, we used cell-specific RNAi to knock down *cha-1*. We expressed double-stranded RNA corresponding to *cha-1* from the *ceh-24* promoter to knock down *cha-1*. Indeed, *cha-1* RNAi in *ceh-24*-expressing cells led to a flipping defect (*Figure 4A*). Because the only cholinergic cells known to express *ceh-24* are SMD, SIB, and SIA, *cha-1* RNAi in these cells likely caused the phenotype. To identify which of the three types of sublateral neurons require *cha-1* for flipping, we performed a rescue mosaic analysis. We used an extrachromosomal array expressing the RNAi construct plus a fluorescent mKate2 marker that co-expressed with the RNAi construct. The array was transmitted to and expressed in only a subset of *ceh-24*-expressing cells, generating a mosaic expression of the *cha-1* RNAi transgene in a subset of *ceh-24*-expressing cells. We cultured and filmed 370 individuals inside microfluidic compartments and scored flipping behavior during L1 sleep. Afterwards, we increased the imaging resolution to check the transgene expression and manually selected those worms that had a clear expression of the array in only a subset of the sublateral neurons. For these, we determined their flip rate. We found one single individual that expressed the RNAi construct only in the SIA neurons and this individual was flipping defective. Also flipping defective were three individuals that expressed the RNAi construct in both SIA and SMD neurons, whereas individuals that carried the transgene only in the SMDs, were not flipping defective. Also, expression in the SIBs was not associated with a flipping defect (*Figure 4A*). The mosaic analysis suggested that *cha-1* expression in the SIA neurons but not in the other neurons is crucial for flipping, yet the numbers of individuals that had specific expression patterns that could be obtained from the mosaic analysis were small despite the large number of animals tested. To confirm the requirement of *cha-1* in the SIA neurons we used another transgene that drove *cha-1* RNAi only in SIA using the *ceh-17* promoter. *ceh-17* also expresses in the ALA neuron but this neuron is not cholinergic, which should result in a SIA-specific knockdown of *cha-1* (*Duerr et al., 2008*; *Pereira et al., 2015*; *Pujol et al., 2000*; *Suo and Ishiura, 2013*). Again, we measured flipping as before. Flipping was greatly reduced by SIA-specific *cha-1* RNAi (*Figure 4B*). To further confirm a role of the SIAs in flipping, we laser-ablated one or two of the four SIAs. We did not ablate all four SIAs, because this appeared to cause pleiotropic effects that likely were unspecific and were probably caused by large doses of laser irradiation. As expected, ablation of some of the SIAs led to a reduction of flipping (*Figure 4C*). To test whether the SIAs are also required for other movements, we assayed for the capacity of SIA::*cha-1RNAi* and *ceh-17(-)* worms to escape from an indentation. Like *ceh-24(-)* worms, both types of SIA-defective worms had problems crawling out of the indentation (*Figure 4D*). We also tried to rescue the flipping defect in *ceh-24(-)* by expression of *ceh-24* from the *ceh-17* promoter, which expresses in the SIAs but not in other sublateral neurons. This transgene rescued neither the process outgrowth nor the flipping defect (*Figure 4—figure supplement 1*). Perhaps, the *ceh-17* promoter did not recapitulate the proper spatiotemporal expression pattern or expression level required for rescue (*Levin et al., 2012*). Thus, the evidence for a role of *ceh-24* in the SIAs is indirect. The loss-of-function results imply that *cha-1* expression and thus cholinergic function in the sublateral SIA neurons is required for flipping and other types of motion. Our Data are consistent with a model in which the SIAs act as cholinergic motor neurons to execute specific movements.

## SIA neurons activate during flipping

How does the activity of the SIAs generate flipping? We used calcium imaging of the sublateral neurons to investigate their activities during flipping and analyzed the activity of the SIA neurons. We expressed the calcium indicator GCaMP6s, which increases green fluorescence upon calcium binding, from the *ceh-24* promoter and cultured worms in microfluidic compartments to film neural activity using time-lapse microscopy and behavior across the sleep-wake cycle (*Chen et al., 2013*). First, we selected and analyzed short sequences of frames covering 80 s that included a flip in the middle of the movie sequence. Individual sublateral neurons were identified manually in each frame and their fluorescence intensity was quantified. The GCaMP signal in the SMD neurons during flipping was low and we could not extract any signals. SIB neurons showed calcium sensor activation, but the

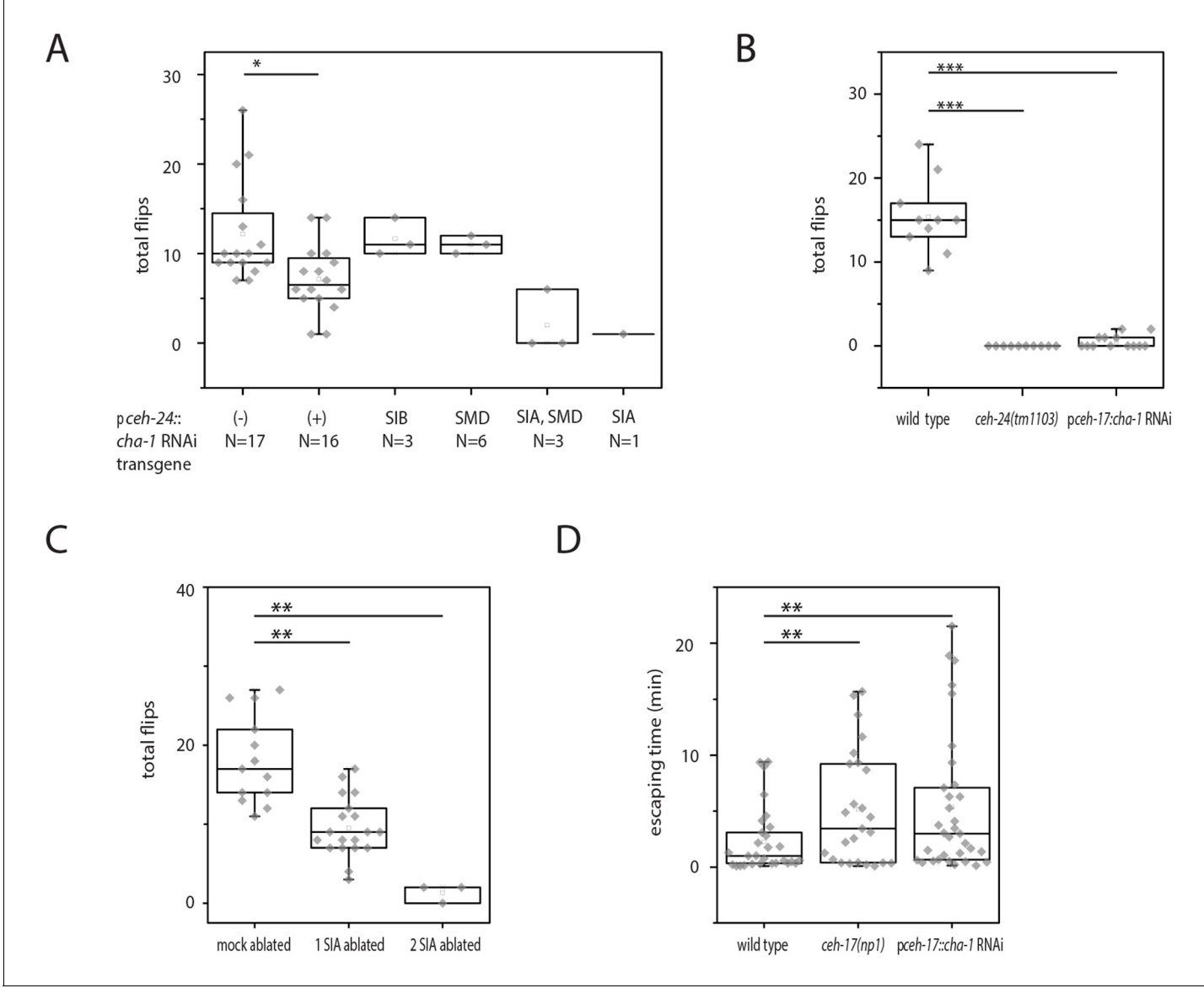

**Figure 4.** Cholinergic function in the SIA neurons is required for flipping. (**A**) Mosaic analysis of *cha-1* RNAi in the sublateral neurons suggests a crucial role for cholinergic transmission from the SIAs: A total of 370 individuals carrying an array expressing double-stranded RNA corresponding to *cha-1* were filmed to score flipping. After filming, individuals with restricted expression in a subset of neurons were selected. (**B**) *cha-1* RNAi in SIA only (driven by the *ceh-17* promoter) confirms a crucial role for cholinergic transmission from these neurons (wild type [N = 10], *ceh-14(tm1103)* [N = 10], p*ceh-17::cha-1* RNAi [N = 13]). (**C**) Ablation of some of the SIAs leads to a flipping defect (mock ablated [N = 13], one SIA neuron ablated [N = 19], two SIA neurons ablated [N = 3]). (**D**) *cha-1* RNAi in SIA leads to a defect in escaping from an indentation (wild type [N = 29], *ceh-17(np1)* [N = 25], p*ceh-17:: cha-1*RNAi [N = 33]). The Kolmogorov Smirnov test was used for all experiments, *** denotes p<0.001, ** denotes p<0.01, * denotes p<0.05.

The following figure supplement is available for figure 4:

**Figure supplement 1.** Lack of rescue of *ceh-24(-)* using *ceh-24(+)* driven by the *ceh-17* promoter.

SIB neurons were close to each other so that signal extraction for individual cells was not possible. Because the SIA neurons are crucial for flipping and because their signal could be extracted with single cell resolution, we focused on the analysis of these neurons. SIA neurons strongly activated during flipping as seen by a rise in calcium indicator fluorescence. While all of the four individual SIA neurons showed activation over baseline, the activation maximum of each of these four neurons

differed, with strong activation of one or two neurons and with less activation of the other SIAs. The individual SIA that activated strongest varied between flips, i.e. different unequal activation patterns of the four SIAs were associated with flipping (*Figure 5A*, individual traces before normalization are shown in *Figure 5—figure supplements 1–2*). During activation, calcium transient maxima typically occurred at the same time for all four SIAs. I.e., rather than activating sequentially, the sublateral neurons appeared to activate simultaneously. Thus, SIA neurons activated all at the same time but the transient maximum of the individual neurons differed. Flipping is not associated with just one particular activation pattern but with several activation patterns. We could not detect differences in timing of activation of the individual SIAs, i.e. the maximum of the calcium transient in each of the four SIAs occurred at the same time. This may be due to slow response times of the indicator or slow acquisition by the EMCCD camera. While subtle differences in timing may contribute to flip generation, our measurements rather suggest that timing differences of SIA activations are less important for flipping. Together, these data suggest a model in which longitudinal rotation is generated through activation of SIA neurons activating muscle quadrants on one side of the worm, which should result in muscle contraction and bending of either the left or the right side (see model Figure 7). Because the worms are confined to a plane, this bending could result in a longitudinal rotation in agreement with a previously proposed theoretical model of flipping that suggested that confinement to a plane is essential for flipping (see discussion and model in Figure 7) (*Tramm et al., 2014*).

## SIA calcium transients occur both during and outside of lethargus

The activation of the sublateral neurons during flipping is consistent with their role in this behavior. But why do flips mostly occur around the time of lethargus? The simplest model would be that the SIA neurons would activate only during lethargus and would thus confine flipping to this developmental stage. However, because the sublateral neurons appear to be also involved in other types of movements these neurons could also be active outside of flipping. We hence looked at sublateral neuron activation also outside of flipping by scoring all activation transients during and outside of lethargus. The neural calcium activity of the SIAs can be described as a baseline activity plus transients above this baseline. We analyzed time-lapse movies that covered the time before, during, and after lethargus and quantified baseline activity and calcium transients by counting their frequencies and maxima and compared calcium transients during three conditions: flipping during lethargus, no flipping during lethargus, and no flipping outside lethargus (*Figure 5B*, individual sample traces before normalization can be found in *Figure 5—source data 1*). Outside of lethargus, many calcium transients were visible. During lethargus, baseline calcium activity was slightly but significantly reduced (*Figure 5—figure supplement 3A*). Calcium transients, however, were still visible (*Figure 5C*, *Figure 5—source data 1* and *Figure 5—figure supplement 3B*). During lethargus, the SIA sublateral neurons activated 0.6 times per minute, and 19% of activation events were associated with a flip. Outside of lethargus, sublateral neurons activated 0.5 times per minute, but only 1% of these events coincided with flipping (*Figure 5C*). Thus, like in many other neurons, baseline calcium activity appeared to be reduced during lethargus in the SIAs. However, calcium transients still occurred during lethargus, consistent with the idea that the SIAs are required for flipping. The majority of SIA calcium transients were not associated with flipping both in and outside of lethargus, but the probability that a transient coincided with a flip was increased in lethargus. Could this be explained by different activation patterns of the SIAs, with only a subset of activity patterns leading to a flip? There was variability in the activation pattern of the SIAs both for transient patterns that were associated with flipping and for transient patterns that were not associated with flipping (*Figure 5—source data 1* and *Figure 5—figure supplement 3B*). Thus, from our calcium data, we could not explain why a certain activation transient pattern would cause a flip and why others would not. Perhaps, other factors determine whether SIA activation results in a flip.

## Optogenetic activation of sublateral neurons can induce flipping during lethargus

Calcium imaging showed that flipping correlated with activation transients in the SIA neurons. But is SIA activation also causative for flipping? To test this idea we activated the SIA sublateral neurons optogenetically by expressing the Channelrhodopsin variant ReaChr, which can be activated by green-to-red light, in the SIA neurons using the *ceh-17* promoter and by providing green light

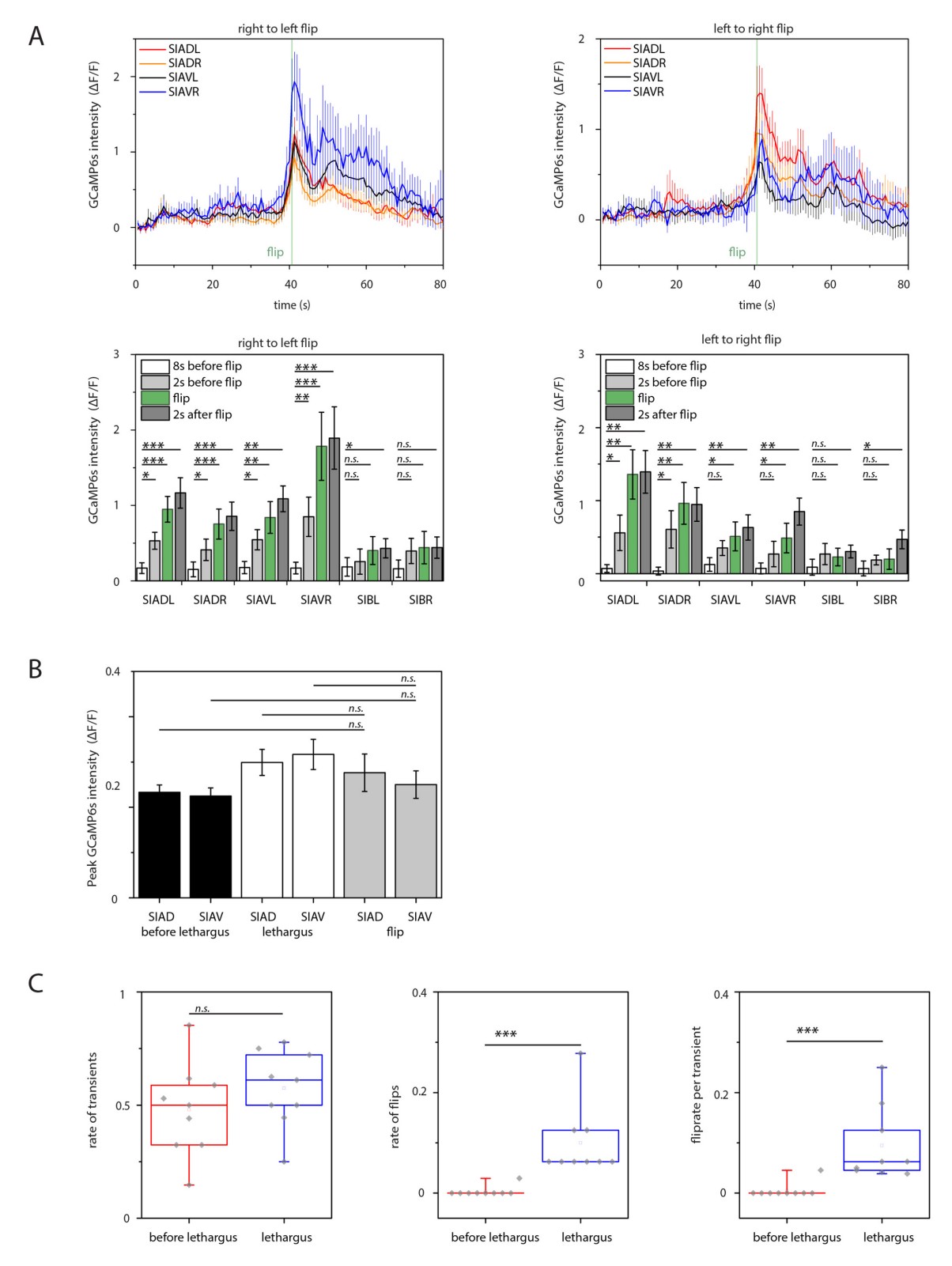

**Figure 5.** Sublateral SIA neurons are active during flipping. (**A**) The SIA neurons activate during flipping: calcium measurements were aligned to the flip. SIA activation and flipping coincided. SIA activation was seen both for flips from the right side to the left side and for flips from the left side to the right side. Histograms show statistical tests for calcium transient maxima (Wilcoxon signed ranks test, right to left flip [N = 14], left to right flip [N = 12]). (**B**) SIA neurons show calcium transients both outside and during lethargus: SIA calcium peaks were compared in three conditions, outside lethargus,

*Figure 5 continued on next page*

*Figure 5 continued*

during lethargus without coincidental flipping, and during flipping (Mann-Whiney U test, nine individual worms were analyzed, numbers of activation transients analyzed were: 17 during flip, 63 during lethargus without flipping, 148 outside lethargus without flipping). (**C**) The probability that an activation of the SIAs coincides with a flip is increased during lethargus. (Mann-Whiney U test, N = 9). *** denotes p<0.001, ** denotes p<0.01, * denotes p<0.05, n.s. denotes p>0.05.

The following source data and figure supplements are available for figure 5:

**Source data 1.** SIA activation transients are highly variable outside of lethargus.

**Figure supplement 1.** Sample calcium transients during left-to-right flipping.

**Figure supplement 2.** Sample calcium transients during right-to-left flipping.

**Figure supplement 3.** SIA baseline calcium sensor signals are reduced during lethargus and heatmap of calcium signal maxima.

stimuli during and outside of lethargus (*Lin et al., 2013*; *Urmersbach et al., 2016*). We cultured worms expressing ReAChr in the SIA neurons in the presence of retinal in microfluidic chambers and activated ReAChr repeatedly every 30 min using a 5 s flash of green/orange light. We filmed the behavior before, during, and after the stimulation using DIC imaging. From these movies, we manually determined the stage of the animal (lethargus or not) and scored whether the worms flipped during ReAChr activation. ReAChr activation resulted in flipping in a fraction of stimulations and the efficiency depended on the behavioral state: Whereas stimulation outside of lethargus led to flipping in 8% of stimulations, during lethargus it led to a flip rate in 17% of stimulations (*Figure 6A*). Thus,

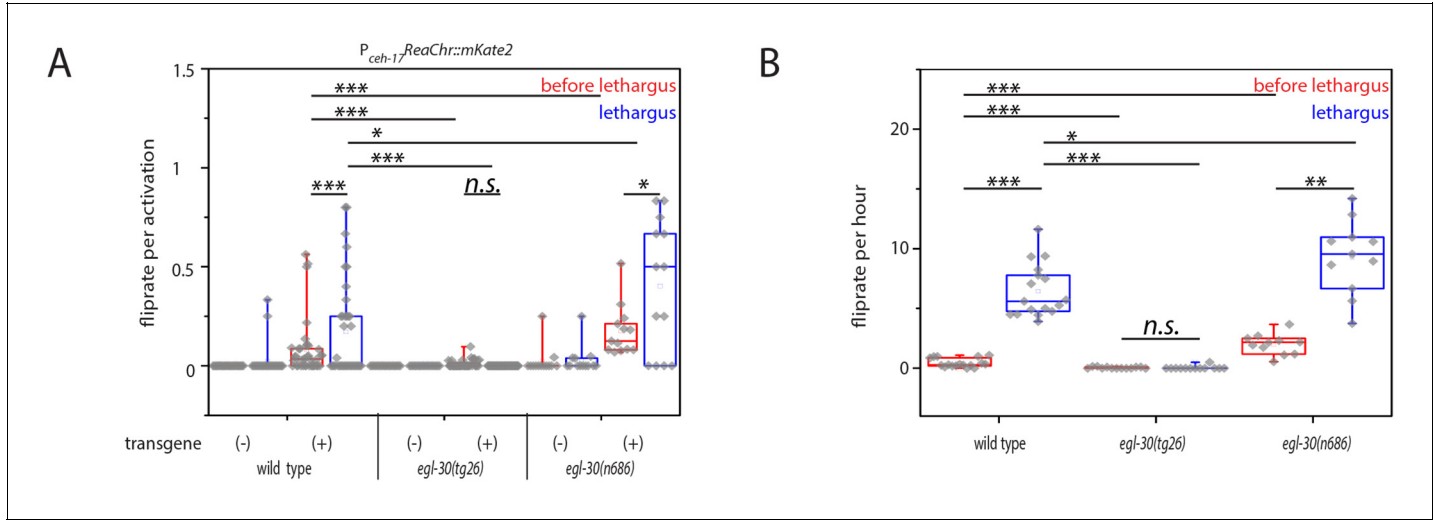

**Figure 6.** Optogenetic activation of SIA neurons can induce flipping with a higher chance during lethargus. (**A**) ReAChR-induced depolarization of the SIAs triggers flipping with a higher chance during lethargus. The SIAs were filmed and were activated every 30 min with yellow light and we scored whether worms flipped during these short movies. SIA activation could induce flipping in and outside of lethargus, but the likelihood of triggering a flip was higher during lethargus. Consistent with the calcium imaging data, the majority of SIA activations did not coincide with flipping. Flipping was impaired in hyperactive *egl-30* mutant worms and increased in hypoactive worms. As controls we used worms of the same strain that did not visibly express the SIA::ReAChr array. Wild type (no transgene N = 25, transgene N = 38), *egl-30(tg26)* (no transgene N = 20, transgene N = 40), *egl-30(n686)* no transgene N = 15, transgene N = 14). (**B**) Endogenous flipping is impaired in hyperactive worms and increased in hypoactive worms. Wild type (N = 17), *egl-30(tg26)* (N = 13), *egl-30(n686)* (N = 11). Kolmogorov-Smirnov test was used for comparisons between genotypes, paired Wilcoxon Signed Ranks test for sleep-wake comparisons, *** denotes p<0.001, ** denotes p<0.01, * denotes p<0.05, n.s. denotes p>0.05.

The following figure supplement is available for figure 6:

**Figure supplement 1.** Heatmap showing variability in extrachromosomal array expression of ReAChr in the SIA neurons.

optogenetics showed that the SIAs can trigger flipping in a fraction of stimulations, consistent with the observation that not every activation of the SIAs coincides with flipping. The probability that sublateral neuron stimulation leads to flipping was increased during lethargus, supporting the hypothesis that lethargus generates a facilitating environment for flipping. Calcium imaging suggested that the strength of SIA activation transients differed in the four individual SIA neurons. However, we have no functional evidence that only unequal SIA activation patterns can lead to flipping. Thus, one might ask whether optogenetic induction of flipping through SIA activation is consistent with the idea that unequal activation of the four SIAs is required for flipping. We hence quantified the expression of ReaChr from the extrachromosomal array in the SIAs and found that the expression levels differed in the individual neurons (*Figure 6—figure supplement 1*). Thus, perhaps, optogenetic activation recapitulated an unequal activation pattern, but this is speculative. Alternatively, the unequal activation pattern of the SIAs may not be crucial for flipping and flipping may also be triggered by equal activation. Targeted illumination approaches could be used in the future to control the activation of the SIAs individually and to test the role of the unequal activation patterns (*Guo et al., 2009*; *Leifer et al., 2011*; *Stirman et al., 2011*).

Why does flipping occur mostly during lethargus and why does it only occur during a small fraction of SIA activations? One hypothetical model consistent with our calcium imaging data is that lethargus generates a facilitating environment during which the likelihood that sublateral activation transients result in flipping is increased. With other words, conditions outside of sublateral neuron activation could either inhibit or promote flipping. Lethargus is characterized by systemic changes in neuronal activity, leading to reduced muscle activity and a relaxed posture. Recently, a biomechanical model was proposed in which a relaxed body posture with fewer body bends facilitates flipping of worms confined to a plane. Consistent with this result, ectopic flipping could be observed in a V-ATPase mutant that showed a relaxed body posture outside of lethargus (*Iwanir et al., 2013*; *Nagy et al., 2014b*; *Schwarz et al., 2011*, *2012*; *Tramm et al., 2014*). We hence set out to directly test the idea that altered activity levels influence the probability that SIA activation leads to flipping.

We investigated mutations that change the activity of neurons and muscles globally. We chose two mutants in *egl-30*, a G alpha q protein required for the activity of excitable cells, because loss- and gain-of-functions exist for this gene and because the role of G alpha q in facilitating synaptic transmission is understood quite well. While null mutants of *egl-30* are lethal, hypomorphic reduction-of-function mutants such as *egl-30(n686)* are slow and have a more relaxed body posture. Hypermorphic gain-of-function mutations in *egl-30* such as *egl-30(tg26)* have increased activity of excitable cells and are behaviorally hyperactive (*Brundage et al., 1996*; *Trent et al., 1983*; *Park and Horvitz, 1986*; *Schwarz and Bringmann, 2013*). We imaged *egl-30(n686)* and *egl-30(tg26)* across the developmental cycle in microfluidic compartments and quantified flipping. Reduction-of-function mutation of *egl-30* led to an increase in flipping rate during lethargus and these mutant worms also showed prominent flipping outside of lethargus. Gain-of-function mutation of *egl-30* led to a decrease of flipping in and outside of lethargus (*Figure 6B*). Next, we optogenetically activated SIA neurons again using ReaChr as described above and measured the efficiency by which neural activation caused flipping. Flipping probability per ReaChr activation was increased in the *egl-30* reduction-of-function mutant, and was decreased in the gain-of-function mutant (*Figure 6A*). Thus, changes in the global activity of excitable cells appear to affect the probability that sublateral neuron activation leads to flipping. While increased behavioral activity is associated with a decreased likelihood that sublateral neuron activation causes flipping, a decreased behavioral activity is associated with an increased probability of flipping. These results are consistent with a model in which flipping is the result of two processes: First, depolarization of SIA neurons, which provides a neuronal basis for triggering this active behavior. And second, a facilitating environment during lethargus that may consist of a relaxed body posture that increases the probability that SIA activation leads to flipping (*Tramm et al., 2014*).

## Discussion

### Flipping is associated with sleep behavior

During lethargus, both motion bouts and quiescence bouts occur. Quiescence bouts are characterized by strongly reduced locomotion, reduced sensory responsiveness, and a more relaxed body

posture. Quiescence bouts are intermitted by motion bouts during which larvae show low levels of locomotion but do not feed. Animals during lethargus motion bouts are more responsive to stimulation compared with animals during a quiescence bout yet have dampened sensory neuron activity and reduced body curvature compared with animals outside of lethargus (*Iwanir et al., 2013*; *Raizen et al., 2008*; *Schwarz et al., 2011*; *Cho and Sternberg, 2014*). Quiescence bouts fulfill the behavioral criteria that define sleep but the exact nature of mobility bouts is not clear yet. At present a parsimonious yet hypothetical interpretation may be that motion bouts present a state that is in between wake and quiescence bout. Most flips emerge from periods in which there is little locomotion and the worm displays a relaxed body posture typically containing only one body bend (*Iwanir et al., 2013*; *Tramm et al., 2014*). Flipping also appears to depend on sleeping behavior, as it is strongly reduced in *aptf-1* mutants that do not show any quiescence bouts during lethargus. The occurrence of flips during periods of low mobility may seem counterintuitive at first, as flipping constitutes a motion and a flip could thus be, by definition, regarded as a motion bout. However, x-y movements of the nose and body typically define a motion bout (*Iwanir et al., 2013*; *Nagy et al., 2014b*). Flipping consists only of a rotation along the longitudinal axis. During this movement, there is almost no x-y movement of the nose or body (see *Figure 1A* for an illustration and compare with [*Tramm et al., 2014*]). A motion threshold cut-off is used in imaging studies to define quiescence bouts. Small movements typically occur also during quiescence bouts. The small motions associated with flipping often fall below the threshold of a motion bout and thus flipping occurs, by this definition, mostly during quiescence bouts (*Nagy et al., 2014b*). Irrespective of the classification into motion bouts and quiescence bouts, flipping occurs preferentially during phases of low mobility.

## A model for sleep-associated flipping

CEH-24 is a conserved NK2 class homeodomain protein that is homologous to mammalian NKX2-1, whose mutation leads to cancer cell survival and progression in humans (*Yamaguchi et al., 2013*). Work on the mammalian homolog showed that this protein is required for organogenesis of the thyroid, lung, pituitary, and ventral forebrain (*Stanfel et al., 2005*; *Kimura et al., 1996*). Here we found that CEH-24 is required for left-right flipping and other types of movements. CEH-24 is expressed and plays a role in the sublateral neurons, where it is required for proper formation of sublateral processes and for the expression of acetylcholine. Thus, the regulation of neural development appears to be a conserved function of CEH-24. These results strongly suggest that *ceh-24* acts in flipping through controlling process morphology and acetylcholine expression in the sublateral neurons. Cell-specific RNAi against the acetylcholine-synthesizing gene *cha-1* showed that acetylcholine expression in SIA neurons is required for flipping and suggests that the SIAs are required for flipping. While the most likely model is that *ceh-24* acts in the SIA neurons to control cholinergic function required for flipping, the evidence for a role of *ceh-24* in the SIAs is indirect, because a rescue of *ceh-24(-)* specifically in the SIAs was not possible. Also, it is possible that other sublateral neurons contribute to flipping in addition to the SIAs. Through loss-of-function experiments we have identified cholinergic function in the SIAs to be crucial for flipping and thus found a crucial role of the SIAs in flipping. These neurons were initially thought to have relatively little output onto other neurons (*White et al., 1986*). Recent reinvestigation of the connectivity of the *C. elegans* nervous system using electron microscopy suggests that additional postsynaptic partners exist (wormwiring.org). Thus, it is formally possible that sublateral neurons act through other neurons to control muscle activity required for flipping. Electron microscopic reanalysis also found synaptic vesicles and presynaptic densities along the sublateral chord, suggesting that the sublateral neurons act as motor neurons and that the sublateral processes directly innervate the four muscle quadrants (*Altun and Hall, 2011*). Separate control of each muscle quadrant has previously been proposed to underlie flipping, but the neural substrates were not known (*Tramm et al., 2014*). The separate innervation of left and right muscle quadrants may not only be an anatomical prerequisite for flipping, but could potentially also underlie other types of movements.

While the sublateral neurons have long been known, their functions have remained unidentified. Here we provide the first functional evidence for a role of the sublateral SIA neurons, which is the control of flipping, a three-dimensional body movement. In addition, the SIAs also have roles in other types of movements, as demonstrated by assaying the ability to escape from an indentation. Consistent with this view, SIA activation transients occur outside of lethargus and most transients do

not coincide with a flip. During lethargus, the baseline calcium activity is reduced, but depolarization transients still occur. Dampening of calcium activity during lethargus has also been reported for other neurons (*Iwanir et al., 2013*; *Schwarz et al., 2011*; *Turek et al., 2013*; *Schwarz and Bringmann, 2013*; *Cho and Sternberg, 2014*; *Choi et al., 2013*). Despite the reduced baseline activity of the SIAs during lethargus, the SIAs still show calcium transients during this stage consistent with the view that these neurons trigger flipping. Flipping likely requires a separate activation of the left and right body wall muscle quadrants. Consistent with this view, SIA neurons activated with different calcium maxima with regard to the individual neuron with one or two of the four neurons activating strongest, and the other SIA neurons activating less. Most of the SIA activation events do not lead to a flip suggesting that flipping depends not only on activation of the SIAs, but also on other factors.

Outside of lethargus, neuronal activity is high and promotes an undulating body posture that forces the larva to lie on its side (*Zhen and Samuel, 2015*). The stronger the dorso-ventral body muscles contract, the stronger the curvature and the more force needs to get exerted to turn the worm about its longitudinal axis. During lethargus, worms engage in sleeping behavior and neuronal and muscle activity is lower and the body muscles are more relaxed (*Iwanir et al., 2013*; *Schwarz et al., 2012*; *Tramm et al., 2014*). If confined to a plane, less force should be necessary to turn the worm about its longitudinal axis if it is only slightly bent compared with an undulating worm that contains more than one bend, as multiple body bends could fix the worm on its side. The same activation strength of the SIAs that may not be sufficient to turn the worm outside of lethargus may become sufficient to turn it during lethargus. As a consequence, flipping would occur more often during lethargus. Thus, the physiological changes that trigger sleep behavior during lethargus would also facilitate flipping. This would explain why flipping is associated with sleeping behavior during lethargus. For example, imagine a sleeping larva that is concavely bent on the ventral side. It could activate SIA neurons leading to a contraction of the right dorsal muscle quadrant. Such a posture would be energetically unfavorable because now the strongest muscle contraction would occur on the right dorsal side, but the dorsal side is convexly bent. A more energetically favorable configuration would be that the dorsal side would be concavely bent. To transition to this energetically more favorable configuration the muscle contraction could either move tail and nose to the opposite side or the worm could rotate about its longitudinal axis to achieve a more bend dorsal side. The first option is impaired due to the confinement of the worm to a plane. Thus, the transition to the posture in which the more contracted side also is the concavely bend side rather occurs through longitudinal rotation (*Figure 7*, [*Tramm et al., 2014*]). This behavior does not occur during waking behavior outside of lethargus, because the worm is fixed by its undulating body posture, preventing longitudinal rotation. In summary, hypothetically, flipping could be possible only during lethargus for two reasons: first, because the sublateral neurons still depolarize, and second because the posture is relaxed. This model would not need a complicated choreography in which different sublateral neurons activate sequentially to achieve flipping. While a timed choreography would result in more efficient flipping, we did not find evidence for such a model in our calcium imaging experiments (*Tramm et al., 2014*). The biomechanical facilitation and the lack of a sequential choreography could together explain why SIA neuron activation results in flipping in only a small faction of events. Together, these data support the hypothesis that both, activity of the SIAs and a relaxed body posture, are required for flipping.

What are the functions of left-right turning during lethargus? We could not find evidence for a role of flipping in molting. The shedding of the cuticle occurs several minutes after the worm has resumed pumping and locomotion activity. Vigorous movements that only slow down once the worm has shed its old cuticle coincide with cuticle shedding. This rather suggests that cuticle shedding is an active process that is triggered not by flipping but by vigorous movements after the period of behavioral quiescence. By contrast, flipping rather appears to be a gentle movement. What then may the function of flipping be? Flipping leads to small changes in the posture, but does not constitute locomotion. Brief muscle activations or changes in the body posture may stimulate metabolic processes in otherwise quiescent muscles, but this idea is purely speculative.

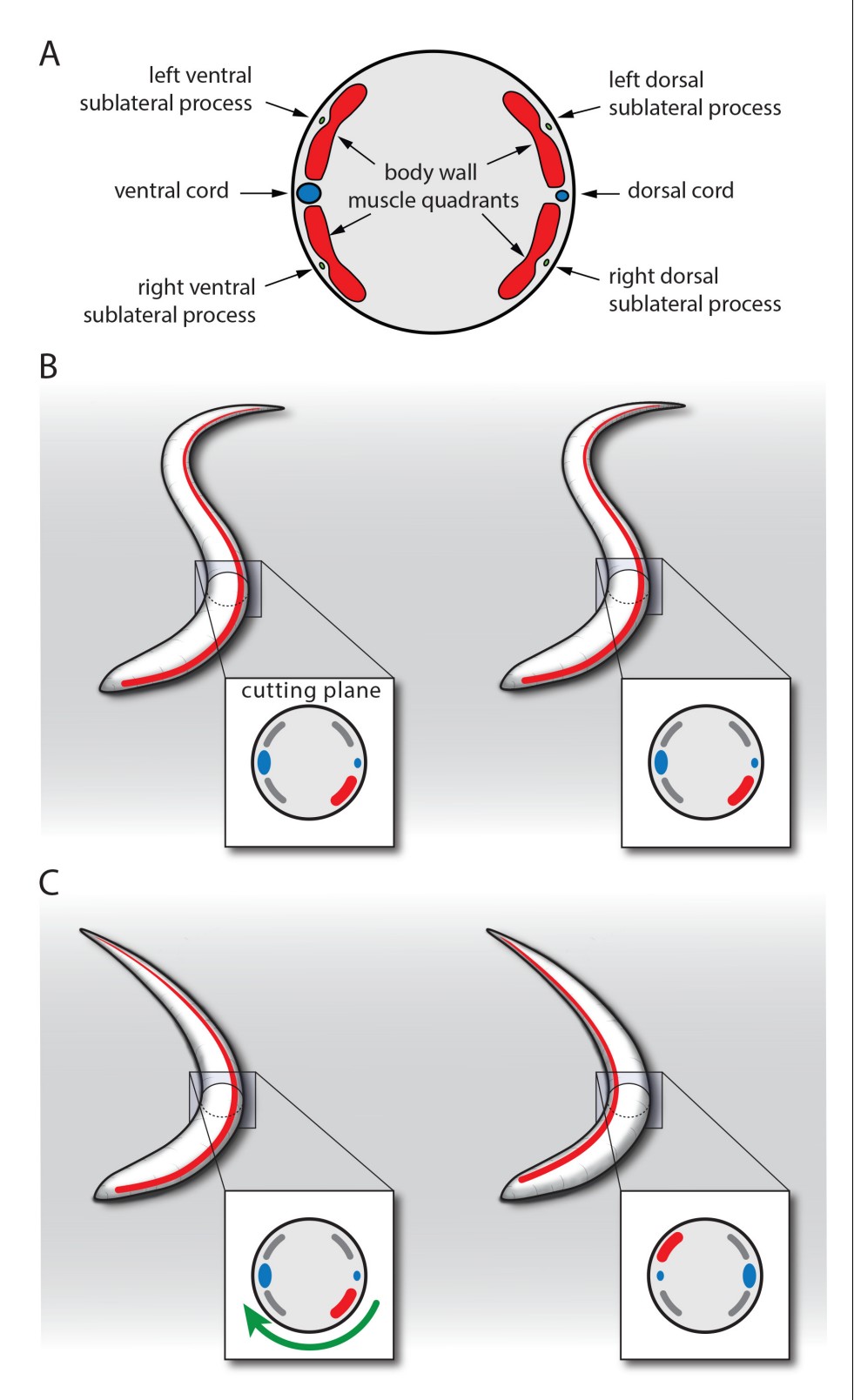

**Figure 7.** Model for SIA neuron-controlled flipping. (**A**) Anatomy of the innervation of body wall muscles by sublateral neurons. Cross-section through *C. elegans* showing the ventral and dorsal cords that control undulating 2D movement. Body wall muscle is organized into four quadrants that run along the long axis of the worm. Along the four body wall muscle quadrants run the sublateral processes, which each emerge from one sublateral SIA neuron (SIAVL, SIAVR, SIADL, and SIADR). Thus, the separate innervation of the four body wall muscle quadrants by the four sublateral SIA processes

*Figure 7 continued on next page*

*Figure 7 continued*

provides an ideal anatomical basis for controlling three-dimensional movements. (**B–C**) Model for SIA-induced flipping: B) In a worm that is outside of lethargus, the body wall muscles are under tension and the SIA neurons may act to control specific movements. Flipping is prohibited because the worm is confined to the plane by the undulating body posture. (**C**) During lethargus, sleep behavior occurs and the overall activity of neurons and muscles is reduced and the worm assumes a less-bent posture. Activation of the SIAs may trigger flipping in some cases in which the musculature on the convex side contracts. This favors a bent posture with the side of the contraction occurring on the concave side. To transition to this posture, the worm rotates 180 degrees around its longitudinal axis. Sublateral SIA neurons likely act as motor neurons that control this three-dimensional motion through separate innervation of the four body wall muscle quadrants.

## Analysis of *ceh-24* identifies sublateral neurons to be required for movements

We initially set out to understand the molecular and cellular basis of flipping. But the low frequency of SIA neuron activation that result in flipping and the strong activity of these neurons outside of flipping suggest that the major role of the SIA neurons is not in flipping. We found that the ability of *ceh-24* mutant worms to crawl was almost normal. Crawling out of an indentation, however, was strongly impaired, suggesting that sublateral neurons are required for locomotion, maybe especially important for specific types of movements. The culture of *C. elegans* on plane agar surfaces has led to strong progress in the understanding of undulating locomotion. Natural environments, such as the rotting biological material on which *C. elegans* is typically found in the wild require three-dimensional navigation and locomotion (*Kiontke and Sudhaus, 2006*). Recent work is starting to establish assays for the study of such types of locomotion. One example is burrowing, with a downward tendency, using magnetic cues for orientation (*Beron et al., 2015*; *Vidal-Gadea et al., 2015*). Another is nictation, a dispersal behavior, in which dauer larvae stand on their tails displaying undulating body movements trying to attach to a larger animal in order to get dispersed and potentially reach new food sources (*Lee et al., 2012*). It would be interesting to test the role of the sublateral neurons, and in particular the role of the SIAs, in these and other types of movements. Because *ceh-24(-)* does not appear to have defects in sensory function and because *cha-1* RNAi in the SIAs cause a motion defect it seems most likely that *ceh-24* and the SIAs play a role in motor function rather than in sensory function. How could sensing of a three-dimensional environment be coupled to navigating through it? The CEP neurons sense the mechanical properties of the surrounding on which *C. elegans* crawls (*Kindt et al., 2007*; *Kang et al., 2010*; *Sawin et al., 2000*). There are four CEP neurons in the head, each with one sensory cilium projecting into the nose. The sensory cilia are positioned so that two of them are located on the ventral side and two of them are located on the dorsal side. In both the ventral and in the dorsal neuron pair one neuron is located on the left and one neuron is located on the right side. Thus, mechanical stimuli could be sensed in all quadrants of the nose. Intriguingly, the CEP neurons have direct connections to the SIA neurons with each CEP connecting to only one SIA. The connectivity directly matches the quadrant in which the CEP cilium is located to the quadrant the sublateral SIA process is located (*White et al., 1986*). Thus, these anatomical connections could allow a direct control of three-dimensional sensing with three-dimensional movement. In summary, we identify the molecular and neuronal bases for flipping in *C. elegans*, and this may be the starting point for also studying other types of motion.

## Materials and methods

### Worm maintenance and strains

*C. elegans* was grown on Nematode Growth Medium (NGM) agarose plates seeded with *E. coli* OP50 and were kept at 25°C (*Brenner, 1974*). The following strains and alleles were used:

**N2:** wild type
**PD4588:** *ceh-24(cc539) V.*
**HBR1000:** *ceh-24(tm1103)V.*
**IB16:** *ceh-17(np1)I.*
**HBR352:** *egl-30(tg26)I.* (Created from CG21 by backcrossing 2x into wild type to remove the him background)

**MT1434:** *egl-30(n686)I.*
**LX929:** *vsIs48[unc-17::GFP].*
**VN411:** *vnEx128[ceh-17::cha-1sense,ceh-17::cha-1antisense, lin-44::gfp, pBluescript].*
**VN412:** *vnEx129[ceh-17::cha-1sense,ceh-17::cha-1antisense, lin-44::gfp, pBluescript].*
**HBR1077:** *goeIs247[pceh-24::GCaMP6s::mKate2-unc-54–3'utr, unc-119(+)].*
**HBR1094:** *ceh-24(tm1103)V; goeIs247[pceh-24::GCaMP6s::mKate2-unc-54–3'utr, unc-119(+)].*
**HBR1130:** *vsIs48[unc-17::GFP]; goeIs253[ceh-24::ReaChr::mKate2-unc-54–3'utr,unc-119(+)].*
**HBR1131:** *ceh-24(tm1103)V; vsIs48[unc-17::GFP]; goeIs253[ceh-24::ReaChr::mKate2-unc-54–3'utr, unc-119(+)].*
**HBR1231:** *goeEx470[pceh-24::mGFP::unc-54–3'utr, unc-119(+), pceh-24::cha-1RNAi::unc-54–3'utr, unc-119(+), pceh-24::cha-1RNAirev::unc-54–3'utr, unc-119(+)].*
**HBR1265:** *unc-119(ed3) III; goeEx501[pceh-24::ceh-24gene::SL2-mKate2-unc-54–3'utr, unc-119(+), [pceh-24::mGFP::unc-54–3'utr, unc-119(+)].*
**HBR1567:** *ceh-17(np1)I; vsIs48[unc-17::GFP]; goeIs253[ceh-24::ReaChr::mKate2-unc-54–3'utr,unc-119(+)].*
**HBR1568:** *egl-30(tg26)); goeEx464[pceh-17::ReaChr::mKate2-unc-54–3'utr, unc-119(+)].*
**HBR1569:** *egl-30(n686) I; goeEx464[pceh-17::ReaChr::mKate2-unc-54–3'utr, unc-119(+)].*
**HBR1800:** *unc-119(ed3)III;goeEx683[pceh-17::ceh-24gene::SL2-mKate2-unc-54–3´utr, unc-119(+)]; ceh-24 (tm1103)V.*

The deletion allele *ceh-24(tm1103)* was backcrossed 10 times against N2 to generate HBR1000. All transgenic insertions were backcrossed at least two times against N2 to remove the *unc-119(-)* background. The following primers were used to detect the deletion in *ceh-24(tm1103)* using a three primer PCR:
Primer 1 – CCA GGT AGA CTT CCA GGC AA
Primer 2 – ACG ACG GAA AAG AAG AGT CCT
Primer 3 – GGG GTG AGC TTC CAT CTT CA

## Molecular biology and transgenic strain generation

All constructs were cloned using the Multisite Gateway system (Invitrogen, Carlsbad, CA) into pCG150 (*Merritt and Seydoux, 2010*). All constructs obtained from LR reactions were sequenced for verification. Sequences used for *cha-1* RNAi were described previously (*Suo and Ishiura, 2013*). The following Gateway entry clones were constructed for this study:

### attL4-pceh-24-attR1
Contains 2850 nucleotides directly upstream of and excluding the ATG start codon of F55B12.1, Wormbase Version WS255,
http://www.wormbase.org/db/get?name=WBGene00000447;class=Gene.
Sequence correctness was verified by Sanger sequencing.

### attL4-pceh-17-attR1
Contains 1190 nucleotides directly upstream of and excluding the start codon of ATG of D1007.1, Wormbase Version WS255,
http://www.wormbase.org/db/get?name=WBGene00000440;class=Gene.
Sequence correctness was verified by Sanger sequencing.

*attL1-SL1-GCaMP6s-gpd-2/SL2-attL2*: contains the GCaMP6s coding region expression optimized for *C. elegans* and intronized (three introns) (*Redemann et al., 2011*). The construct contains an SL1 before and an SL2 site after the GCaMP6s coding sequence to allow construction of an operon with mKate2. The first 34 amino acids of GCaMP6s were omitted similar as done before for GCaMP3 (*Schwarz et al., 2011*).

### *attL1-ceh-24-attL2:* cDNA of *ceh-24*
900 bp of the single splice form of F55B12.1, Wormbase Version WS255, http://www.wormbase.org/db/get?name=WBGene00000447;class=Gene. Sequence correctness was verified by Sanger sequencing.
*attR2-mKate2-unc-54–3'utr-attR2*: expression-optimized mKate2 gene (*Schwarz et al., 2011*)

attL1-ReaChr-attL2: expression-optimizmized ReaChr gene (*Lin et al., 2013*; *Urmersbach et al., 2016*)

The following Gateway final constructs were used:
pceh-24:: ReaChr::mKate2-unc-54–3'UTR,unc-119(+)
pceh-24::GCaMP6s::mkate2-unc-54–3' UTR, unc-119(+)
pceh-17::ReaChr::mKate2-unc-54–3' UTR, unc-119(+)
pceh-24::cha-1RNAi::unc-54–3' UTR, unc-119(+)
pceh-24::cha-1RNAirev::unc-54–3' UTR, unc-119(+)
pceh-24::ceh-24::SL2-mKate2-unc-54–3' UTR, unc-119(+)
pceh-17::ceh-24::SL2-mKate2-unc-54–3' UTR, unc-119(+)

## Transformation

We generated transgenic strains by microparticle bombardment or by microinjection using *unc-119 (ed3)* rescue as a selection marker (*Wilm et al., 1999*; *Praitis et al., 2001*). Injection concentration was 60 ng/µl for all constructs.

## Genetic screening

For genetic screening we used time-lapse movies from ~40 strains that we had from a previous screen for sleep mutants. These mutants were initially selected because we suspected that they had defective sleep behavior, but they turned out to have normal sleep as judged by the time-lapse movies (*Turek et al., 2013*). For the sleep screen, we had filmed between two and four individuals in microfluidic compartments made from agarose hydrogel (190 × 190 µm, 10 µm deep) using DIC microscopy and a time lapse protocol with one frame each five seconds. Movies were then scrolled through quickly for the presence of flips as identified by the side of the developing gonad. The examiner was blinded for the underlying mutations during scoring of flipping. For subsequent experiments, the examiner was not blinded any longer. For PD4588, no flips were detected in four out of four animals filmed. While the number of strains screened was quite low, we did not continue screening but rather focused on characterizing *ceh-24*.

## Long-term imaging

All long-term imaging experiments were carried out using agarose microchamber imaging (*Bringmann, 2011*; *Turek et al., 2015*). Nose tracking was performed manually. Calcium imaging was performed similarly as described before using GCaMP3.35 and co-expression of mKate2 as an expression control (*Schwarz et al., 2011*, *2012*; *Turek et al., 2013*; *Schwarz and Bringmann, 2013*). For calcium imaging, we used an Andor iXon (512 × 512 pixels) EMCCD camera and LED illumination (CoolLed) using standard GFP and Texas Red filter sets (Chroma). Exposure times were in the range of 5–20 ms and allowed imaging of moving worms without blurring. The EMCCD camera triggered the LED through a TTL 'fire' signal to illuminate only during exposure. LED intensity was in the range of 15–30%. EM gain was between 0 and 100. All calcium-imaging experiments were done in agarose microchambers. Typically, 15–50 individuals were cultured in individual microchambers that were in close vicinity. Animals were filmed by taking a z-stack in a continuous mode (*Figure 4A*) or in a 'burst mode', which means that a short movie of z stacks was taken every 15 min (*Figure 4B, C,D*). Each burst consisted of 50 z-stacks (with each stack consisting of 19 planes with 1 µm resolution) with a frame rate of 20 pictures / second. Individual compartments were repeatedly visited by using an automatic stage (Prior Proscan2/3) set to low acceleration speeds. Before each fluorescent measurement, we took a brief DIC movie to assess the developmental stage and behavioral state. Larvae that showed pharyngeal pumping were scored as being in the wake state. We manually selected a z plane for each neuron we wanted to quantify (*Figure 4A*) and projected fluorescence pictures to extract left and right neurons using Andor IQ Software (*Figure 4B,C,D*). Fluorescence signals were cut out using semi-automated homemade Matlab routines, in which neurons were manually identified and were then cut out automatically. We scored frequency and intensity of the peaks by counting them and measuring their maximum size (*Figure 4B,C,D*). The baseline was determined separately for each activation transient by averaging five frames before each transient. We noticed that the baseline was slightly reduced in the SIA neurons by approximately 16% during sleep compared with wake. We computed the activation difference over baseline. To allow for a better

comparison of transients during sleep and wake we normalized transient strength in wake by using the baseline during sleep for each individual animal. Long-term imaging was similarly described previously (*Turek et al., 2016*). All imaging experiments were carried out at 20°C. Quiescence bout analysis was carried out similar to described (*Iwanir et al., 2013*; *Nagy et al., 2014b*). Movement below a threshold of 1.65 µm/s was used to define a quiescence bout and movement above this threshold was used to define a motion bout. The length and frequency of the quiescence bouts were extracted using a Matlab routine.

## Optogenetics

ReaChr experiments were performed inside agarose microchambers as described (*Turek et al., 2016*; *Lin et al., 2013*). We grew hermaphrodite mother worms on a medium that was supplemented with 0.2 mM all trans Retinal (Sigma). We then placed the eggs from these mothers together with food into microchambers without any further Retinal supplementation. We stimulated ReaChr with an LED of 585 nm with about 0.1 mW/mm (*Cirelli and Tononi, 2008*) as measured with a light voltmeter. We took 10 frames with a rate of 2/s before stimulation, i.e. filmed for 5 s, then, we stimulated ReaChr for 10 s, during which we continued with image acquisition, and then finally we filmed the worms again after stimulation for 5 s. This protocol was repeated for the same individual worm every 30 min. Using an automated stage, we imaged 10–50 individuals in one overnight round of experiments. The experiment was controlled by Andor iQ software. For mosaic analyses, after ReaChr stimulation and imaging, we added 25 mM levamisole onto the chambers to immobilize the worms. We then took high-resolution (1000x) stacks of the mKate2 expression signal to identify neurons expressing mKate2.

## Spinning disc imaging

We used spinning disc imaging with an Andor Revolution spinning disc system using a 488 nm laser and a 565 nm laser, a Yokogawa X1 spinning disc head, a 100x oil objective and an iXon EMCCD camera. Z stacks were taken and a maximum intensity projection calculated using iQ software as described in (*Schwarz et al., 2011*).

## Laser ablation

Laser ablation were as described previously. Briefly, we used a 355 nm laser focused to a near-diffraction limited spot (Rapp Opto, DPSL-355/14, direct coupling). All ablations were performed at 1000x magnification in a strain expressing mKate2 in the sublateral neurons. Neurons were identified and ablated in late embryos using mKate2 fluorescence without using anesthetics. We verified the successful ablation of neurons using mKate2 fluorescence, and worms were discarded when they showed unspecific laser damage or when the neurons of interest were not successfully ablated. We used mock-ablated worms that were treated like ablated worms except that they were not irradiated with the laser (*Turek et al., 2013*).

## Adult assays

Movement measurements on a plane surface: To characterize movement on a plane surface, young adult worms were placed onto an NGM (Nematode Growth Plate) seeded with OP50 bacteria. Animals were recorded using a camera mounted onto a Leica MZ15 Stereomicroscope using dia illumination for 7 min with 1 frame per second. Backwards and forwards movement was scored manually and also the position of the nose was tracked manually to determine movement speeds.

Response to mechanical stimulation: Responsiveness to gentle touch was tested as previously described (*Chalfie et al., 2014*). Worms were touched briefly at their head behind the pharynx using an eyelash glued to a holding stick. An individual was scored as responding to mechanical stimulation if it stopped movement or initiated a backward movement. Each individual was tested ten times and a total of 30 individuals were tested per condition.

Escape from an indentation: Indentations were box shaped with the following dimensions: 700 µm x 700 µm (xy) and 65 µm deep. Indentations were cast from 3% agarose in S-Basal by using a PDMS stamp as described (*Bringmann, 2011*). Fresh indentations were cast for each individual experiment. Unlike in the long-term culture experiments, chambers were not sealed with a glass coverslip but were left open. No food was added to the chambers. Individual young adult worms were

picked into the indentation. The sample was then placed onto a time-lapse microscope and the behavior of the animal was recorded immediately with two frames per second at 100x magnification using bright field microscopy. The time point at which a worm had left the indentation was scored manually and was defined as the first point in time when the entire body of the worm was outside the indentation.

## Statistics

Statistical tests used were Mann Whitney U test, Kolmogorov-Smirnov test, Wilcoxon Signed Ranks test, or Student's t-test using Origin software. The specific test used is described in the figure legends. Error bars are SEM throughout.

## Acknowledgements

We thank Ines Lewandrowski for taking the screen movies, Michael Pilot for scoring screen movies for flipping, Maurice Haedrich, for help with data processing, the CGC (funded by the NIH Office of Research Infrastructure Programs P40 OD010440), the MITANI Lab (through the National Bio-Resource Project of the MEXT Japan), Satoshi Suo for strains, Hartmut Sebesse for help with figure drawings, and David Biron for helpful discussions regarding flipping during sleep.

## Additional information

### Funding

| Funder | Grant reference number | Author |
|---|---|---|
| Max-Planck-Gesellschaft | Max Planck Research Group | Henrik Bringmann |
| Max-Planck-Gesellschaft | Open-access funding | Henrik Bringmann |

The funders had no role in study design, data collection and interpretation, or the decision to submit the work for publication.

### Author contributions

JS, Data curation, Formal analysis, Investigation, Methodology, Writing—review and editing; HB, Conceptualization, Supervision, Funding acquisition, Writing—original draft

### Author ORCIDs

Henrik Bringmann, http://orcid.org/0000-0002-7689-8617

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
