## [Decision Letter]

[Editors’ note: a previous version of this study was rejected after peer review, but the authors submitted for reconsideration. The first decision letter after peer review is shown below.]

Thank you for submitting your work entitled "The NK2 homeobox gene *ceh-24* is required for sublateral motor neurons to control left-right turning during sleep" for consideration by *eLife*. Your article has been favorably evaluated by a Senior Editor and three reviewers, one of whom is a member of our Board of Reviewing Editors.

Our decision has been reached after consultation between the reviewers. Based on these discussions and the individual reviews that are summarized at the end of this letter, we regret to inform you that your work can not be considered in its present from at *eLife*. In general, the reviewers expressed interest in your reported findings in principle, but identified a substantial number of significant problems. If you are able to address these points, we encourage you to submit this manuscript as a new submission to *eLife*.

Combined reviews:

1) Lethargus is not sleep, it's a developmental stage.

*C. elegans* sleep in short bouts for roughly 25% of the time during the developmental stage called lethargus; they are active in motion bouts for roughly 75% of lethargus. Animals do not flip in sleep bouts, they flip during lethargus motion bouts. It is inappropriate to call lethargus motion bouts "sleep" and misleading to say that flipping occurs during sleep. Animals behaviorally respond to external stimulation during motion bouts as well as they respond in adult and inter-molt stages. Arousal thresholds are an accepted and critical metric for defining any behavior as sleep. The reviewer realizes that calcium transients in sensory neurons are reduced during lethargus motion bouts, but that is not sufficient to call these bouts sleep. The manuscript starts by equating motion bouts with sleep in the title and continues with this conflation through the entire manuscript. At a minimum, the authors must rewrite the entire manuscript as "flipping is common during the developmental stage called lethargus".

2) There is no insight provided into what underlies "state-specific" behavior.

The manuscript states that "Little is known about how a behavior becomes state-specific" and that "connectivity of circuits is thought to be unaltered during different behavioral states, activities within these circuits differ between states." Actually, there is a great deal known about the "state-specific" behavior. One obvious case is locomotion during hunger versus satiety. While the manuscript shows SIA neurons are required for flipping during lethargus, problematically there is no insight into what makes flipping a "state-specific" behavior. The lethargus "state change" that allows flipping is functionally downstream of SIA neurons.

The Abstract states that "SIA-induced flipping is sleep-specific because these neurons are not shut down during sleep and because a global dampening of excitable cells facilitate this behavior" The first conclusion is inconsistent given a contradictory statement in the Methods section and there is virtually no evidence presented that supports the last conclusion. They do note that *egl-30* animals can be induced to flip more readily as adults by SIA optical stimulation. But, EGL-30 is a broadly expressed G-protein found in many tissues. It has diverse roles and these observations are not sufficient to support the manuscript's claims. Much more work would be required to delineate "state-specific" behavior. For example, where EGL-30 is acting in this behavior? Does it really regulate global dampening of excitable cells during lethargus? Without this extensive analysis, there is no real insight into "state-specific" behavior.

3) Significant changes are required in presentation of results.

Figure panels with single values or columns are unacceptable. They do not help the reader understand the actual variation in the results/determinations. The new standard in the field is to show individual determinations whenever possible. If the results presented are the% of animals in an entire population, the data should be presented as a small table. If multiple replicates/individuals are averaged, then show the actual data for each replicate/individual in the figure. The only exception is when the figure would be over-whelmingly complicated.

Also, the manuscript does not make clear to the reader how many biological versus technical replicates are included in all panels/tables/experiments. Specifically, biological replicates are animals from different parents assayed on a different day. Technical replicates are animals reared together and assayed on the same day. This fine-grained information must be included in a supplemental table.

Raw data before normalization must be provided.

In Figure 2, orientation of animals and the number/identity/location of cell bodies is not obvious to readers, especially in 2B and 2C. Cartoon/diagrams indicating cell bodies and outlines of animals would be helpful. And, the red font is challenging to read; at a minimum this should be bold or larger font size. The left hand panels in 2C and D are superfluous; a single small table in the figure would suffice to present the data in the bar graphs.

Methods need more details. Several examples follow. First, to state that a clone "contains 2850 nucleotides upstream of *ceh-24*" is not sufficiently precise to guide readers. Second, which cDNA of *ceh-24*? Third, what concentrations were injected for transgenesis? Fourth, in which experiments were researchers blinded as to genotype/treatment/conditions? Fifth, more details on the "escape from indentation assay are needed". These are just examples.

4) "These results rather suggest that ceh-24 mutant worms have a general defect in crawling through a three-dimensional environment." This statement of *ceh-2*4 function is not supported. *ceh-24* is expressed in multiple neuron types, including in several types of motor neurons. It is conceivable that *ceh-24* animals have locomotory defects in both 2-dimensional and 3-dimentional environments. However, the locomotory defects were not characterized by the authors. It is critical to do so. In addition, it remains possible that defects in mechanical sensation reduced the ability to crawl out of an indentation, because mechanical sensation may play a role in localizing the indentation. Similarly, the quantification of sleep in *ceh-24*- animals, based on locomotion, is cursory and should be more detailed to reach the standards of the field.

5) The focus of action of *ceh-24* is likely in SIA, but the evidence is only circumstantial. The authors should examine (and properly quantify) whether the effect of *ceh-24* on *unc-17* is restricted to SIA (i.e. other neurons such as SMDs and SIBs are not affected). This would strengthen their circumstantial evidence and is an essential experiment to do. Additionally, the authors need to show rescue using the *ceh-17* prom to drive *ceh-24*. I don’t consider this essential if *ceh-24* affect *unc-17* only in SIA, but it does become essential if *ceh-24* does affect other SMD and SIB, too.

Generally, the presentation of the RNAi experiments is concerning. First, the *ceh-24* promoter is not only expressed in motor neurons. Therefore, the conclusion that "Indeed, *cha-1* RNAi in sublateral neurons led to a flipping defect (Figure 3)." is not supported. Second, the mosaic analysis only identified one animal that expressed RNAi only in SIA and three animals that expressed RNAi in SIA and SMD. These are really small sample sizes to draw a conclusion upon.

The RNAi experiments do not demonstrate the sufficiency. A cell-specific rescue is needed to convincingly establish the role of SIA.

6) The presentation of the calcium imaging experiments is difficult to understand or evaluate. "These neurons activated appeared asymmetric with strong activation of one or two of the four SIA neurons and with less activation of the other SIAs". "Asymmetric" does not appear to be accurate in this context. "Rather than activating sequentially, the sublateral neurons appeared to activate simultaneously" "Spontaneously" rather than "simultaneously" should be more precise. "We could not detect differences in timing of activation of the individual sublaterals" What does "timing" mean?

More importantly, given the complexity of the calcium traces that the authors described, some or all of the calcium traces should be included in the manuscript.

7) "With the exception of the sleep-active neuron RIS, all other neurons tested so far had reduced activity during sleep". Does the statement describe results from this study or others? If other neurons were tested in this study, the results need to be shown to support the key conclusion of the paper.

8) The optogenetic experiments are confusing and a bit surprising. The authors showed that SIA neurons usually activate "asymmetrically" and proposed that the "asymmetrical activation" triggers flipping. How did they generate the "asymmetric activation" of SIA neurons in the optogenetics experiments?

---

## [Author Response]

[Editors’ note: the author responses to the first round of peer review follow.]

*Combined reviews:*

1) Lethargus is not sleep, it's a developmental stage.

*C. elegans sleep in short bouts for roughly 25% of the time during the developmental stage called lethargus; they are active in motion bouts for roughly 75% of lethargus. Animals do not flip in sleep bouts, they flip during lethargus motion bouts. It is inappropriate to call lethargus motion bouts "sleep" and misleading to say that flipping occurs during sleep. Animals behaviorally respond to external stimulation during motion bouts as well as they respond in adult and inter-molt stages. Arousal thresholds are an accepted and critical metric for defining any behavior as sleep. The reviewer realizes that calcium transients in sensory neurons are reduced during lethargus motion bouts, but that is not sufficient to call these bouts sleep. The manuscript starts by equating motion bouts with sleep in the title and continues with this conflation through the entire manuscript. At a minimum, the authors must rewrite the entire manuscript as "flipping is common during the developmental stage called lethargus".*

We agree with the reviewer that behavior during lethargus can be classified as either quiescence bout or motion bout (as defined by work from the Biron lab, Iwanir, 2013). Dependent on the method used, quiescence bouts can occupy more than 50% of lethargus (Nagy, Raizen, Biron, 2014). The Biron lab also published that the majority of flips do not occur during motion bouts but during quiescence bouts (Tramm et al. 2014). Specifically, it was found that a strong bias exists for flips to occur during postures that contain only one body bend. A relaxed body posture with reduced bending has been shown to be typical for *C. elegans* sleep (Iwanir et al. 2013 and Schwarz 2013). With other words, during a quiescence bout, the animal does not show locomotion and they display a relaxed body posture that typically contains only one body bend, whereas animals during locomotion typically contain more body bends. The occurrence of a single body bend was found to be greatly correlated with flipping suggesting a functional requirement of this posture in flipping and led to a biomechanical model for flipping (Tramm et al. 2014). Thus, a strong link between quiescence bouts and flipping exists in the published literature.

We have based our manuscript on this literature data. Tramm et al. 2014 has clearly stated that flipping occurs mostly “during” sleep: “Flips were only observed when the animals were quiescent and only during a small fraction of the quiescence bouts.” We did not find any literature data that would show that flipping occurs only during motion bouts. The reviewers’ statement that flipping occurs “during lethargus” is consistent with the literature data, but we have used the term “during sleep” in our initial manuscript because this is more specific and is in line with the literature data. Nevertheless, this appears to be an important point and we have done two experiments to corroborate the finding that flipping occurs during motion bouts/sleep:

A) We have measured mobility before each flip and we found that the majority of flips occurred during periods of reduced mobility. This is consistent with the results from the Biron lab that stated that flips occur mostly “during” quiescence bouts (Tramm et al. 2014). This experiment does not prove causality between low levels of motion and flipping, however. Thus, in a second experiment, we looked at the effect of sleep loss on flipping:

B) We have looked at flipping in a mutant that does not show sleep behavior (i.e. a mutant that lacks quiescence bouts). We chose the *apft-1* deletion because it is the strongest sleep mutant available. It completely lacks quiescence bouts during lethargus and has no other reported phenotypes (Turek et al. 2013). *aptf-1(-)* worms showed greatly reduced flipping consisting with the idea that reduced mobility facilitates flipping.

Together with the literature data, it appears that low mobility coincides with the majority of flips and it also appears that flips do not specifically occur during motion bouts.

We were wondering how the impression emerged that “flipping occurs during motion bouts” and we think that there are some semantic considerations that may have played a role: It is clear that flipping is a movement. If sleep were defined as the complete absence of movement, then flipping would – by definition – interrupt the quiescence bout and – again by definition – occur during a motion bout.

However, flipping does not cause much x-y movements, because flipping consists only of a rotation along the longitudinal axis. See, for instance, the worm in Figure 1. x-y movements have been used so far to define motion bouts and quiescence bouts. Movement below a certain threshold defines a quiescence bout – and there is some residual movement even during quiescence bouts (Iwanir et al. 2013 and Nagy, Raizen, Biron, 2014).

The reconciliation could be that flips constitute a small motion that may fall below the detection threshold for a motion bout.

Finally, we also discussed this point with David Biron, who has first published the distinction between quiescence bout and motion bout (Iwanir et al. 2013) and has also published first that most flips occur during sleep (Tramm et al. 2014). Based on this conversation we think that the way we present our paper is along the line of the published literature and consistent with all evidence that we are aware of. Please feel free to involve Dr. Biron in the discussion if you would like to take it further.

In conclusion, we respectfully differ from the reviewers’ comment that “flips occur during motion bouts”. Based on the results from the Biron lab and our own results that are consistent with the results from the Biron lab it appears that most flipping emerges from quiescence bouts and that flipping depends on sleep behavior.

It appears that this point was brought up because of a limited understanding of flipping in the field. Thus, we feel that there is the need to discuss and describe the relationship between flipping and sleep more clearly in the paper and to add additional data to clarify this point. We have done the following three things to improve the paper in this regard and to address this point:

A) Explanation of “lethargus”, “sleep”, and “bouts”:

We have rewritten the entire manuscript and have made clearer what “lethargus” is and have added an explanation of what “quiescence bouts” and “motion bouts” are and how they relate to sleep and to flipping. We now use the term “lethargus” when we refer to the developmental stage.

B) Addition of data on flipping during quiescence bout / sleep mutant:

We have also added data to corroborate results from the Biron lab that most flips occur during periods of low mobility. Our results are consistent with the findings from the Biron lab. Also, we added data from a sleep mutant (*aptf-1(-)*), which does not show any quiescence bouts and which has strongly reduced flipping.

Together this describes the link between sleeping behavior and flipping.

C) Discussion of the semantics

Whether flips occur “during” a quiescence bout or whether they “interrupt” a quiescence bout partly is a semantic question. But what is clear is that there is a strong link between sleeping behavior and flipping and thus we feel that it is fair to state in this manuscript that worms flip “during” sleep or that flipping is “associated” with sleep and the new version of our manuscript is written along these lines. We have added a section to the Discussion to explain these issues and we have rephrased the manuscript accordingly.

2) There is no insight provided into what underlies "state-specific" behavior.

*The manuscript states that "Little is known about how a behavior becomes state-specific" and that "connectivity of circuits is thought to be unaltered during different behavioral states, activities within these circuits differ between states." Actually, there is a great deal known about the "state-specific" behavior. One obvious case is locomotion during hunger versus satiety. While the manuscript shows SIA neurons are required for flipping during lethargus, problematically there is no insight into what makes flipping a "state-specific" behavior. The lethargus "state change" that allows flipping is functionally downstream of SIA neurons.*

*The Abstract states that "SIA-induced flipping is sleep-specific because these neurons are not shut down during sleep and because a global dampening of excitable cells facilitate this behavior" The first conclusion is inconsistent given a contradictory statement in the Methods section and there is virtually no evidence presented that supports the last conclusion. They do note that egl-30 animals can be induced to flip more readily as adults by SIA optical stimulation. But, EGL-30 is a broadly expressed G-protein found in many tissues. It has diverse roles and these observations are not sufficient to support the manuscript's claims. Much more work would be required to delineate "state-specific" behavior. For example, where EGL-30 is acting in this behavior? Does it really regulate global dampening of excitable cells during lethargus? Without this extensive analysis, there is no real insight into "state-specific" behavior.*

In the new version of our manuscript we have a stronger focus on explaining the relationship between sleep and flipping. We think that we provide an exciting novel motor system that is required for flipping associated with sleep and our current manuscript focuses on describing this system. In the course of rewriting the manuscript we removed the focus of “state specificity” and have changed all parts of the manuscript accordingly. We have also removed the introductory paragraph on state specificity.

Calcium experiments:

The presentation of the calcium imaging data was greatly reworked and you can find details on these experiments in the respective section below. The bottom line is that there is a small decrease in baseline activity of GCaMP signals in the SIAs but activation transients still are present in lethargus.

*egl-30* experiments:

The goal of the *egl-30* experiment was to generate a systemic effect on excitable cells to link our discovery of the SIA function to a previously published biomechanical model of flipping (Tramm et al.. 2014). The model states that a more relaxed body posture facilitates flipping. We wanted to test whether decreasing activity levels increases the chance that the activation of the SIAs causes flipping. To this end we used a partial loss-of-function mutation of a broadly expressed regulator of neural function to generate a global state of reduced neural activity, which may coarsely mimic some aspects of the changes in the activity of excitable cells during sleep. What exactly constitutes sleep at the level of the physiology of every single neuron is currently unknown. Hence it is

not technically possible to reconstitute all aspects of neural dampening at present. We found that *egl-30* hypomorphic mutation increases optogenetically-induced flipping.

The reviewer suggested that we find out where *egl-30* acts and whether it controls neural dampening during sleep. We feel that the focus of this manuscript should not be on the role of *egl-30*. We have worked intensively on *egl-30* in lethargus and found it extremely difficult to identify a site of action of *egl-30* in the control of sleep during lethargus (Schwarz et al. 2013). We do not try to prove any specific role of *egl-30* in any specific cell type in this manuscript. We also did not try to test a role of *egl-30* in lethargus neural dampening. The goal was simply to broadly affect animal behavior and test if reducing *egl-30* function would increase flipping chance. The observation that we can even induce flipping in wake *egl-30* loss-of-function animals supports a biomechanical model that was described by the David Biron lab (Tramm et al. 2014) and our goal was to link our findings on the molecular and neural substrates of flipping to this previously published model.

In summary we have three things to address this comment:

A) We removed the focus of “state specificity” to “molecular and neural substrates” in this manuscript.

B) We reworked the presentation of calcium data, see data presentation part below.

C) We de-emphasized the interpretations of the *egl-30* experiments

We feel that an extensive additional analysis of *egl-30* would not provide further important insights into state specificity. We have kept the *egl-30* experiment in the present manuscript to link our findings on the molecular and neural substrates of flipping to a previously published biomechanical model of flipping, which we then discuss. If the reviewer feels strongly about the *egl-30* experiments we could also take them out.

3) Significant changes are required in presentation of results.

*Figure panels with single values or columns are unacceptable. They do not help the reader understand the actual variation in the results/determinations. The new standard in the field is to show individual determinations whenever possible. If the results presented are the% of animals in an entire population, the data should be presented as a small table. If multiple replicates/individuals are averaged, then show the actual data for each replicate/individual in the figure. The only exception is when the figure would be over-whelmingly complicated.*

*Also, the manuscript does not make clear to the reader how many biological versus technical replicates are included in all panels/tables/experiments. Specifically, biological replicates are animals from different parents assayed on a different day. Technical replicates are animals reared together and assayed on the same day. This fine-grained information must be included in a supplemental table.*

Raw data before normalization must be provided.

*In Figure 2, orientation of animals and the number/identity/location of cell bodies is not obvious to readers, especially in 2B and 2C. Cartoon/diagrams indicating cell bodies and outlines of animals would be helpful. And, the red font is challenging to read; at a minimum this should be bold or larger font size. The left hand panels in 2C and D are superfluous; a single small table in the figure would suffice to present the data in the bar graphs.*

*Methods need more details. Several examples follow. First, to state that a clone "contains 2850 nucleotides upstream of ceh-24" is not sufficiently precise to guide readers. Second, which cDNA of ceh-24? Third, what concentrations were injected for transgenesis? Fourth, in which experiments were researchers blinded as to genotype/treatment/conditions? Fifth, more details on the "escape from indentation assay are needed". These are just examples.*

We reworked all data figures to provide follow the suggestions by the reviewer. We provide data on individuals for all experiments.We provide information on replicates in the “transparent reporting” file. We provided raw data and several example traces for the calcium data. We added cartoons to the figures to help orient the reader. We added more details to the Methods section as requested.

*4) "These results rather suggest that ceh-24 mutant worms have a general defect in crawling through a three-dimensional environment." This statement of ceh-24 function is not supported. ceh-24 is expressed in multiple neuron types, including in several types of motor neurons. It is conceivable that ceh-24 animals have locomotory defects in both 2-dimensional and 3-dimentional environments. However, the locomotory defects were not characterized by the authors. It is critical to do so. In addition, it remains possible that defects in mechanical sensation reduced the ability to crawl out of an indentation, because mechanical sensation may play a role in localizing the indentation. Similarly, the quantification of sleep in ceh-24- animals, based on locomotion, is cursory and should be more detailed to reach the standards of the field.*

We agree with the reviewers that, because *ceh-24* controls cholinergic function in motor neurons, it is conceivable that there are additional roles of these neurons in motion. This makes the sublateral neurons extremely interesting to study in the future in the contexts of different types of movement. We have not seen defects of movement on the plate, and we have quantified locomotion speed plus backward and forward movements, and have added this information to the supplement. Superficially, these worms crawl normally on the plate. This is consistent with work from Andy Fire’s lab that could not detect any phenotype (Harfe and Fire 1998). This is what prompted us to challenge the *ceh-24* mutant by using a structured surface and this is where we found a clear phenotype. We cannot exclude that there are more subtle phenotypes in 2D locomotion. We have rewritten the text so that we do not discriminate between 2D vs. 3D any longer. While a lot more could be done to understand the role of the sublaterals in other types of motion, we feel that this would be beyond the scope of this manuscript. What is interesting about *ceh-24* is that it is involved in some types of movement that appear more complex than crawling on a plate and this makes the sublateral neurons interesting to study in the future.

*ceh-24* is expressed in motorneurons but not in sensory neurons. Consistent with this, work from Andy Fires lab has not found a defect of the mutant in sensory assays (Harfe and Fire 1998). We have tested the response of *ceh-24(-)* to gentle touch as suggested by the reviewers and could not find any phenotype.

We have added this information to the supplement. Also, in our initial manuscript we included an experiment in which we looked at escape from the indentation in *ceh-17::cha-1RNAi* worms that specifically have defects in cholinergic function in the SIAs. This SIA – specific manipulation recapitulated the escape phenotype pointing towards a role of the SIA motor neurons as opposed to a sensory function. From these experiments it seems very unlikely that *ceh-24*/SIA acts as a sensory system but rather suggests a motor neuron function. We have added this argument to the Discussion of our new manuscript.

We have also improved the analysis of the sleep phenotype based on locomotion. To link to the discussion on quiescence bouts we have measured quiescence bouts and have added this information to the supplement. Bout analysis suggests that locomotion quiescence is normal in *ceh-24(-)*. We make a strong point in our manuscript that *ceh-24* defines cholinergic function of the SIAs and this role of *ceh-24* is not specific to sleep, but also present outside of this behavior. Thus, we feel that a further characterization of sleep behavior in *ceh-24(-)* would not lead

to further insights into flipping.

*5) The focus of action of ceh-24 is likely in SIA, but the evidence is only circumstantial. The authors should examine (and properly quantify) whether the effect of ceh-24 on unc-17 is restricted to SIA (i.e. other neurons such as SMDs and SIBs are not affected). This would strengthen their circumstantial evidence and is an essential experiment to do. Additionally, the authors need to show rescue using the ceh-17 prom to drive ceh-24. I don’t consider this essential if ceh-24 affect unc-17 only in SIA, but it does become essential if ceh-24 does affect other SMD and SIB, too.*

*Generally, the presentation of the RNAi experiments is concerning. First, the ceh-24 promoter is not only expressed in motor neurons. Therefore, the conclusion that "Indeed, cha-1 RNAi in sublateral neurons led to a flipping defect (Figure 3)." is not supported. Second, the mosaic analysis only identified one animal that expressed RNAi only in SIA and three animals that expressed RNAi in SIA and SMD. These are really small sample sizes to draw a conclusion upon.*

*The RNAi experiments do not demonstrate the sufficiency. A cell-specific rescue is needed to convincingly establish the role of SIA.*

Expression of ceh-24:

The *ceh-24* is expressed in SMD, SIB, and SIA as well as in muscle cells (Harfe and Fire 1998, Kennerdell et al. 2009). All *ceh-24* expressing neurons have sublateral processes. Because the three types of sublateral neurons are the only cholinergic cells that express *ceh-24*, it is likely that the effect observed by *cha-1* RNAi stems from these cells, i.e. sublateral motor neurons. In addition, we showed a specific effect for the SIAs by *ceh-17*-driven *cha-1* RNAi (which is specific to SIAs, see below). We rewrote this part to make it clearer to the reader that *ceh-24* neurons are sublateral neurons.

Quantification of cholinergic marker

We looked at and quantified *unc-17* expression in SIB and SMD in wild type and *ceh-24* mutants as requested. SIB and SMD also expressed the cholinergic marker and marker expression was reduced in *ceh-24(-)*.

Rescue:

We also cloned the *ceh-24* coding DNA with the *ceh-17* promoter and tested for rescue of the *ceh-24* phenotype as requested. The transgene rescues the process outgrowth defect of the SIAs only partially. Consistent with this, this transgene also did not rescue flipping.

A rescue experiment is nice if it works. For a rescue experiment to work the timing and strength of expression may be crucial. In this case a rescue of the process outgrowth is likely required for SIA function. Process outgrowth occurs during a specific time during development. It is thus likely that rescue gene expression would need to recapitulate the endogenous timing of expression and likely also the endogenous amount of expression. Expression in the same neuron in the adult may not be enough. We checked in the expression databases for the embryonic expression pattern of *ceh-24* and *ceh-17* as deposited by the Yanai lab (Levin 2012). The timing of these factors differ, which may speculatively explain why the rescue is partial. At present, it seems to be impossible to do this experiment, because of a lack of a promoter that recapitulated the proper spatiotemporal and level of expression of *ceh-24* only in the SIAs.

The rescue experiment could have shown that the SIAs are sufficient among the sublateral neurons. Our goal was to identify neurons required for flipping.

Requirement can best be tested with loss-of-function experiments. We did a loss- of-function experiment using *cha-1* RNAi. First, we did this experiment with the *ceh-24* promoter and used a mosaic analysis. Because the number of experimental animals for RNAi in SIA was low, we also did *cha-1* RNAi from the *ceh-17* promoter to knock down *cha-1* specifically in the SIAs. This second experiment provided a sufficient number of animals that did not flip showing clearly that the SIAs are required for flipping. The statement written by the reviewers make us wonder whether this second, SIA specific RNAi experiment, which was present in our initial manuscript, has been overlooked. We agree that it would be nice to directly demonstrate a role of *ceh-24* in the SIAs. But we do not think that a rescue is “essential” to prove the role of the SIAs in flipping, because loss-of-function evidence is available.

To further corroborate the role of the SIAs in flipping we did an additional loss-of- function experiment: We ablated one or two of the four SIAs using a UV laser beam and found that ablation of some of the SIAs caused reduced flipping (ablation of three or all four SIAs made the animals unspecifically sick – maybe due to the large amount of laser irradiation, and we decided not to use extend the ablation to all cells for this reason).

In summary, we think that we provide evidence that *ceh-24* is required for flipping, that *ceh-24* is required for sublateral neuron cholinergic function, and that cholinergic function in SIA is required for flipping. We have clearly demonstrated a role of the SIAs in flipping, which make a function of *ceh-24* in the SIA likely, even though the evidence is circumstantial. We cannot directly show a role of *ceh-24* in the SIAs, but we also don’t think that this is essential for defining a role of the SIAs in flipping. We have changed the presentation of the data to make this clear and we discuss this in the Discussion part of the new manuscript.

*6) The presentation of the calcium imaging experiments is difficult to understand or evaluate. "These neurons activated appeared asymmetric with strong activation of one or two of the four SIA neurons and with less activation of the other SIAs". "Asymmetric" does not appear to be accurate in this context. "Rather than activating sequentially, the sublateral neurons appeared to activate simultaneously" "Spontaneously" rather than "simultaneously" should be more precise. "We could not detect differences in timing of activation of the individual sublaterals" What does "timing" mean?*

*More importantly, given the complexity of the calcium traces that the authors described, some or all of the calcium traces should be included in the manuscript.*

We rewrote this section to describe the observations better. We removed the term “asymmetrical” and tried to describe the data better. “Timing” is also described in more detail now. We also added several calcium traces as supplementary information and hope that now the complexity of these data is documented.

*7) "With the exception of the sleep-active neuron RIS, all other neurons tested so far had reduced activity during sleep". Does the statement describe results from this study or others? If other neurons were tested in this study, the results need to be shown to support the key conclusion of the paper.*

We referred here to literature data. The neurons and references are now clearly stated in the text and references are added.

*8) The optogenetic experiments are confusing and a bit surprising. The authors showed that SIA neurons usually activate "asymmetrically" and proposed that the "asymmetrical activation" triggers flipping. How did they generate the "asymmetric activation" of SIA neurons in the optogenetics experiments?*

We quantified expression of ReaChr from the *ceh-17* promoter and it appears that expression levels in the four individual SIAs are unequal. Speculatively, this could cause also an unequal activation of the four SIAs. We cannot exclude that symmetrical activation of the SIAs could also cause flipping. To clearly generate unequal activation patterns and to test hypotheses as to the precise activation pattern of these neurons they would need to be illuminated with high spatial and temporal control. This, however, is difficult because the cell membranes (containing Channelrhodopsins) of the SIAs are touching each other and also, such experiments require a highly sophisticated setup that we currently don’t have access to. The unequal expression pattern from the *ceh-17* promoter is consistent with the idea that unequal activation is required but also other interpretations are possible. We have added this information to the manuscript and also mention the possibility that symmetrical activation cannot be excluded to also trigger flipping. We feel, however, that causal testing of activation patterns would be a challenging story on its own and beyond the scope of this manuscript.